# The cholesterol uptake regulator PCSK9 promotes and is a therapeutic target in APC/KRAS-mutant colorectal cancer

Chi Chun Wong [1✉], Jian-Lin Wu[2], Fenfen Ji[1], Wei Kang [3], Xiqing Bian[2], Huarong Chen [1], Lam-Shing Chan[1], Simson Tsz Yat Luk[1], Samuel Tong[1], Jiaying Xu[1], Qiming Zhou[1], Dabin Liu [1], Hao Su[1], Hongyan Gou[1], Alvin Ho-Kwan Cheung[3], Ka Fai To [3], Zongwei Cai [4], Jerry W. Shay [5] & Jun Yu [1✉]

Therapeutic targeting of *KRAS*-mutant colorectal cancer (CRC) is an unmet need. Here, we show that Proprotein Convertase Subtilisin/Kexin type 9 (PSCK9) promotes APC/KRAS-mutant CRC and is a therapeutic target. Using CRC patient cohorts, isogenic cell lines and transgenic mice, we identify that de novo cholesterol biosynthesis is induced in *APC/KRAS* mutant CRC, accompanied by increased geranylgeranyl diphosphate (GGPP)—a metabolite necessary for KRAS activation. PCSK9 is the top up-regulated cholesterol-related gene. PCSK9 depletion represses *APC/KRAS*-mutant CRC cell growth in vitro and in vivo, whereas PCSK9 overexpression induces oncogenesis. Mechanistically, PCSK9 reduces cholesterol uptake but induces cholesterol de novo biosynthesis and GGPP accumulation. GGPP is a pivotal metabolite downstream of PCSK9 by activating KRAS/MEK/ERK signaling. PCSK9 inhibitors suppress growth of APC/KRAS-mutant CRC cells, organoids and xenografts, especially in combination with simvastatin. PCSK9 overexpression predicts poor survival of *APC/KRAS*-mutant CRC patients. Together, cholesterol homeostasis regulator PCSK9 promotes *APC/KRAS*-mutant CRC via GGPP-KRAS/MEK/ERK axis and is a therapeutic target.

[1] Institute of Digestive Disease, Department of Medicine and Therapeutics, State Key Laboratory of Digestive Disease, Li Ka Shing Institute of Health Sciences, The Chinese University of Hong Kong, Hong Kong SAR, China. [2] State Key Laboratory of Quality Research in Chinese Medicine, Macau Institute for Applied Research in Medicine and Health, Macau University of Science and Technology, Macau, China. [3] Department of Anatomical and Cellular Pathology, The Chinese University of Hong Kong, Hong Kong SAR, China. [4] State Key Laboratory of Environmental and Biological Analysis, Department of Chemistry, Hong Kong Baptist University, Hong Kong, China. [5] Department of Cell Biology, University of Texas Southwestern Medical Center, Dallas, TX, USA. ✉email: chichunwong@cuhk.edu.hk; junyu@cuhk.edu.hk

CRC is the third most common cancer worldwide[1]. CRC involves a cascade of genetic mutations, with APC and KRAS being the major players. *APC* loss-of-function mutations is one of earliest events in colorectal carcinogenesis and ~85% of CRC harbor defects in APC[2]. *KRAS* is mutated in 45% of CRC and it drives CRC progression. *APC* loss and mutant *KRAS* synergistically induces CRC and drug resistance[3,4]. Novel therapies for *APC/KRAS*-mutant CRC is urgently needed.

Cholesterol is an essential lipid component and it plays multifaceted roles in normal physiology and tumorigenesis[5]. Essentiality of cholesterol biosynthesis to cancer growth are exemplified by synthetic lethality screens showing that CRISPR/Cas9 knock-out of cholesterol biosynthesis genes is incompatible with cell viability[6]. However, pharmacological blockade with statins showed limited efficacy in CRC[7]. Cholesterol biosynthesis pathway generates intermediary metabolites that engage in signaling cascades[8], however, it is unclear which of these metabolites are essential for *APC/KRAS*-mutant CRC cells.

Since *APC/KRAS*-mutant CRC are addicted to metabolic deregulation[9,10], we aim to identify metabolic pathways altered in this significant subset of CRC. Gene set enrichment analysis (GSEA) of RNA-seq datasets from *APC*-mutant, or *APC/KRAS*-mutant CRC in comparison to wildtype counterparts demonstrated that cholesterol metabolism was specifically enriched in *APC/KRAS*-mutant CRC subset. Gene profiling of isogenic cell lines confirmed that cholesterol metabolism was synergistically induced by mutant *KRAS* plus *APC* alterations. We identified in CRC with *APC/KRAS*-mutations acute induction of proprotein convertase subtilisin/kexin type 9 (PCSK9). Recent work has indicated that PCSK9 might be a therapeutic target for cancer immunotherapy[11], however, its potential role in CRC with respect to *KRAS* mutation remains unexplored. Functional and mechanistic investigations demonstrated that PCSK9 dysregulated cholesterol homeostasis to induce biosynthesis of geranylgeranyl pyrophosphate (GGPP), which activated KRAS/MEK/ERK cascade to promote cell growth. Targeting of PCSK9 in conjunction with statins suppressed growth of *APC/KRAS*-mutant CRC, suggesting that PCSK9 is a therapeutic target for a large subset of patients with CRC.

## Results

### *APC/KRAS*-mutant CRC is enriched in the cholesterol biosynthesis pathway

To uncover the metabolic pathways aberrantly regulated in *APC/KRAS*-mutant CRC, we utilized the RNA-seq datasets from the Cancer Genome Atlas (TCGA) colon and rectal cancer (COAD-READ) cohorts. CRC arises from the sequential mutations of *APC* and *KRAS*, we thus compared *APC*-mutant CRC ($N = 142$) and *APC/KRAS*-mutant CRC ($N = 134$) subsets versus their wildtype counterpart ($N = 50$). Gene set enrichment analysis (GSEA) demonstrated the enrichment of 18 and 24 KEGG pathways (FDR < 0.05), respectively. Overlap of the enriched gene sets revealed 8 unique pathways enriched in *APC/KRAS*-mutant CRC (Fig. 1A). Two of these pathways, Terpenoid Backbone Biosynthesis and Steroid Biosynthesis, are involved in cholesterol biosynthesis and are simultaneously enriched in *APC/KRAS*-mutant CRC (Fig. 1B), suggesting the activation of cholesterol biosynthesis.

### *APC* loss and *KRAS* mutation in colonic cells mediate a shift in cholesterol homeostasis from transcellular uptake to de novo biosynthesis

Cholesterol could be acquired from extracellular source or biosynthesized via de novo cholesterol biosynthesis pathway[5]. To evaluate the contribution of these pathways in *APC/KRAS*-mutant CRC, we utilized 1CT isogenic cells expressing sh*APC* (1CT-A), mutant *KRAS*$^{G12V}$ (1CT-K), or both (1CT-

AK)[12]. We measured uptake of low-density lipoprotein (LDL) cholesterol using BODIPY-labelled LDL as a tracker. The sequential introduction of sh*APC* and *KRAS*$^{G12V}$ into 1CT cells progressively suppressed the cellular uptake of BODIPY-labelled LDL by confocal microscopy (Fig. 1C). Both semi-quantitative (Fig. 1C) and quantitative (Fig. 1D and Fig. S1) analyses of LDL fluorescence by field counting and flow cytometry, respectively, confirming that uptake of LDL in 1CT-AK cells was lower than that of 1CT, 1CT-K, and 1CT-A cells. Consistent with repressed LDL uptake, LDLR was down-regulated in 1CT-A and 1CT-AK cells (Fig. 1E). ICT-AK cells also demonstrated increased *ABCA1* but decreased *NPCL11* mRNA expression, which may contribute to increased cholesterol uptake (Fig. S2). We next evaluated de novo cholesterol biosynthesis with deuterated water (D$_2$O)-based stable isotope labeling[13]. LC-MS/MS analysis of deuterium incorporation into cholesterol revealed the minimal incorporation in 1CT and 1CT-K cells; whereas increased labelling of cholesterol was detected in 1CT-A (3%) and 1CT-AK (4.8%) cells, indicative of increased de novo cholesterol biosynthesis (Fig. 1F). However, the total cholesterol levels remain unchanged in these isogenic cell lines. These results implied sh*APC* plus mutant *KRAS* in colonic cells reduced their dependence on cholesterol uptake, concomitant with an increased reliance on de novo cholesterol biosynthesis. To corroborate this, we analyzed the dependencies of 1CT isogenic cells with or without lipoproteins (80% depletion). 1CT, 1CT-K and 1CT-A cells showed stunted growth in lipoprotein-depleted DMEM, while the cell growth of 1CT-AK cells was maintained (Fig. 1G). Together, *APC* loss plus mutant *KRAS* in colonic cells leads to increased cholesterol biosynthesis and reduced reliance on cellular uptake.

### Increased de novo cholesterol biosynthesis induces accumulation of bioactive isoprenoids in vitro and in vivo

We next evaluated intermediary cholesterol metabolism by LC-MS/MS. We found that 3-hydroxy-3-methylglutaryl-CoA (HMG-CoA) and geranylgeranyl pyrophosphate (GGPP) were induced in 1CT-AK as compared to 1CT cells (Fig. 1H). Total free cholesterol was unchanged. GGPP represents an alternative branch point of the mevalonate pathway, implying that increased flux via de novo cholesterol biosynthesis is directed towards GGPP. We therefore assessed intermediary cholesterol metabolites in *Apc*$^{min/+}$*Kras*$^{G12D/+}$*Villin-Cre* mice. As shown in Fig. 1I, GGPP ($P < 0.01$), geranyl pyrophosphate (GPP) ($P < 0.05$), and farnesyl pyrophosphate (FPP) ($P < 0.05$) were up-regulated in colon tumors compared to adjacent normal tissues (Fig. 1I), implying higher isoprenoid biosynthesis, especially GGPP. We speculated that GGPP might also be up-regulated *Apc*$^{Min/+}$ given its increased level in 1CT-A cells. Concomitant up-regulation of GPP, FPP and GGPP may imply more robust up-regulation of mevalonate pathway in mice as compared to cell line models. GGPP is a metabolite that mediates protein prenylation essential for Ras and Rho activation. We demonstrated that GGPP promoted colony formation of 1CT-K and 1CT-AK cells, implying that it functions as an oncometabolite in a mutant KRAS-dependent fashion (Fig. S3). Collectively, *APC* loss plus mutant *KRAS* promotes the up-regulation of cholesterol biosynthesis and GGPP accumulation in cell lines and mice.

### *APC* loss and *KRAS* mutation in colonic cells induce the cholesterol biosynthesis pathway with PCSK9 as a key target

To unravel the molecular basis of altered cholesterol metabolism, we profiled gene expression in the 1CT isogenic cells with PCR arrays (Human Fatty Acid Metabolism PCR array and Human Lipoprotein Signaling and Cholesterol Metabolism PCR array) for 161 unique genes. The top up-regulated gene in 1CT-AK cells was

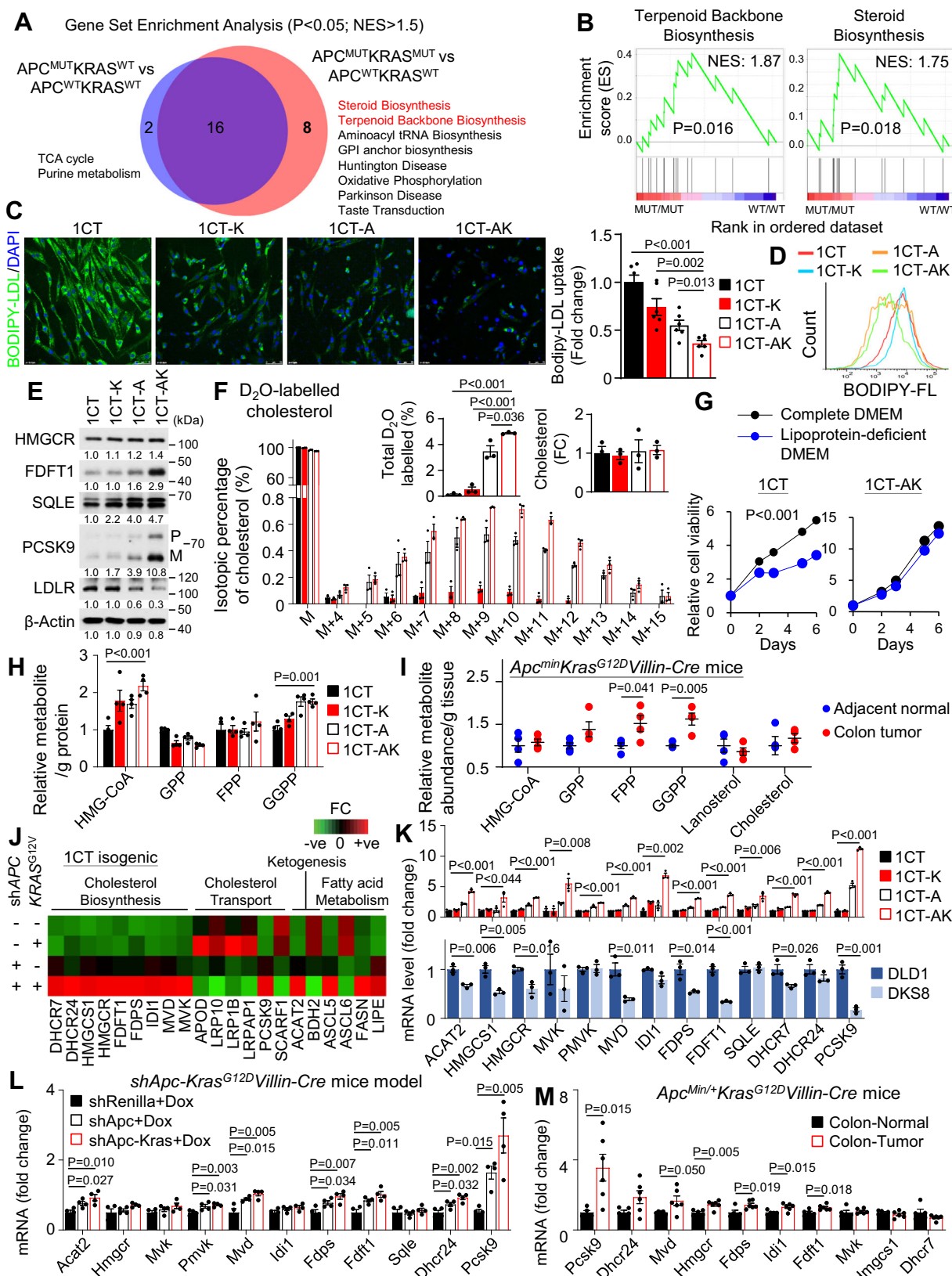

*PCSK9* (>9-fold over 1CT cells), together with up-regulation of cholesterol biosynthesis genes (Fig. 1J), whereas lipoprotein receptor-related proteins were down-regulated. We validated up-regulated genes by qPCR (Fig. 1K). In another isogenic pair, DLD1 (*Apc^Mut^KRAS^G13D/+^*) and its wildtype KRAS counterpart DKS8 (*Apc^Mut^KRAS^WT^*) cells, the loss of mutant *KRAS* allele

reduced mRNA levels of *PCSK9* and cholesterol biosynthesis genes (Fig. 1K). Western blot of 1CT isogenic cells confirmed the up-regulation of PCSK9, FDFT1 and SQLE in 1CT-AK cells (Fig. 1E). We also evaluated the effect of *APC*-loss and *Kras*^G12D^ on cholesterol metabolism-related gene expression in genetic CRC mice models (sh*Renilla*, sh*Apc* and sh*Apc-Kras*^G12D^)

**Fig. 1 Cholesterol homeostasis is shifted from cellular uptake to de novo biosynthesis in *APC/KRAS*-mutant CRC leading to increase geranylgeranyl pyrophosphate (GGPP) levels. A** Gene Set Enrichment Analysis (GSEA) of enriched KEGG pathways in *APC*-mutant (*N* = 142), *APC/KRAS*-mutant CRC (*N* = 134) as compared to wildtype counterparts (*N* = 50) from the Cancer Genome Atlas (TCGA) colon and rectal cancer (COADREAD) cohort. Permutation-based *p*-value. **B** Terpenoid Backbone Biosynthesis pathway and Steroid Biosynthesis pathway are enriched in *APC/KRAS*-mutant CRC cases from TCGA cohort. Permutation based *p*-value. **C** Fluorescence microscopy of cellular uptake of fluorescent labelled LDL (BODIPY-LDL) in 1CT isogenic cell lines (1CT-K: 1CT-*KRAS*$^{G12V}$, 1CT-A: 1CT-sh*APC*, 1CT-AK: 1CT-sh*APC*-*KRAS*$^{G12V}$). Reduced LDL uptake was observed in 1CT-AK cells (*n* = 6). **D** Flow cytometry of intracellular fluorescence after incubation with BODIPY-LDL. **E** Western blot confirmed the up-regulated protein expression of PCSK9 and cholesterol biosynthesis genes in 1CT-AK cells; LDLR was inhibited. **F** D$_2$O stable isotope labeling (48 h) of cholesterol and LC-MS analysis of intracellular cholesterol in 1CT isogenic cells, revealing increased de novo cholesterol biosynthesis in 1CT-A and 1CT-AK cells, while having no effect of total cholesterol level (*n* = 3). **G** Cell viability of 1CT-AK cells were not sensitive to lipoprotein depletion in culture medium. Cell growth curve of 1CT and 1CT-AK cells in complete or lipoprotein-depleted medium (*n* = 3). **H** LC-MS of cholesterol pathway intermediates in 1CT isogenic cells. **I** LC-MS of cholesterol pathway intermediates in CRC and adjacent normal tissues from 7-weeks-old *Apc*$^{min/+}$*Kras*$^{G12D/+}$*Villin-Cre* mice (*n* = 4). **J** PCR array (Human lipoprotein signaling & cholesterol metabolism; and Human fatty acid metabolism) revealed that *KRAS*$^{G12V}$ plus sh*APC* increased mRNA expression of cholesterol biosynthesis genes. **K** qPCR validated the increased mRNA expression of *PCSK9* and cholesterol biosynthesis genes in 1CT-AK cells compared to 1CT, 1CT-K, and 1CT-A cells, whilst knockout of mutant *KRAS* in DLD1 cells (i.e., DKS8) exerted an opposite effect (*n* = 3). **L** PCSK9 and cholesterol biosynthesis genes were up-regulated in the sh*Apc*-*Kras*$^{G12D}$-*Villin-Cre* mice colon (*n* = 4) compared to sh*Ren* controls (*n* = 4) and sh*Apc*-*Villin-Cre* mice (*n* = 4). **M** mRNA levels of *PCSK9* and cholesterol biosynthesis genes in *Apc*$^{Min/+}$*Kras*$^{G12D}$*Villin-Cre* mice tumors (*n* = 6) compared to adjacent normal tissues (*n* = 5). Data shown are means of biological replicates; ± SEM (**C**, **F**–**I**, **K**–**M**). Two-tailed Student's *t*-test (**C**, **F**, **H**, **I**, **K**–**M**). Two-tailed two-way ANOVA (**G**). Source data are provided as a Source Data file.

(GSE67186), showing that *PCSK9* and cholesterol biosynthesis genes were induced in sh*Apc*-*Kras*$^{G12D}$ compared to sh*Renilla* or sh*Apc* mice colon (Fig. 1L). *PCSK9* was also the top up-regulated cholesterol pathway gene in tumors compared to adjacent normal tissues in *Apc*$^{Min/+}$ *Kras*$^{G12D/+}$-*Villin-Cre* mice (Fig. 1M). These data suggest that PCSK9 expression is driven by concurrent *APC* and *KRAS* mutations in CRC.

**APC loss and KRAS mutation up-regulate PCSK9 and cholesterol biosynthesis genes via β-catenin-mediated transcription.** We next probed the mechanism underlying up-regulation of PCSK9 and cholesterol biosynthesis genes. In silico promoter prediction unveiled β-catenin-related (LEF/TCF) binding sites in *PCSK9* promoter (Table S1). β-Catenin is associated with cholesterol biosynthesis gene transcription[14]. Chromatin-immunoprecipitation-PCR indicated that β-catenin binds to *PCSK9* promoter (Fig. S4), and β-catenin knockdown in 1CT-AK cells led to marked decline in the expression of *PCSK9* and cholesterol biosynthesis genes (Fig. S4). Notably, β-catenin knockdown had a greater effect on *PCSK9* expression in 1CT-AK compared to 1CT cells (Fig. S4). Mutant *KRAS* activates β-catenin[3], as evidenced by increased nuclear active β-catenin translocation, TOPflash activity and expression of WNT/β-catenin target genes (Cyclin D1/D2, CD44, EPHB2 and c-Myc) in 1CT-AK cells as compared with 1CT-A cells (Fig. S5). β-catenin-induced thus PCSK9 represents a point of synergy between *APC* and *KRAS* in CRC.

**PCSK9 promotes malignant phenotypes in APC/KRAS-mutant CRC cells.** PCSK9 plays a central role in modulating cholesterol homeostasis by repressing cellular uptake of lipoproteins[11]. Given the acute up-regulation of PCSK9 (5- to 10-fold) upon *APC/KRAS* mutation in cell lines and in mice, we reasoned that PCSK9 drives altered cholesterol homeostasis and its associated growth promoting effect. We silenced PSCK9 in 1CT isogenic cells by siRNAs (Fig. 2A) and demonstrated that, si*PCSK9* dramatically inhibited cell viability and colony formation of 1CT-AK cells, but had no effect on 1CT cells (Fig. 2B, C). PCSK9 knockout in KRAS-mutant SW1116 and LOVO cells (Fig. 2D) reduced cell growth and colony formation (Fig. 2E). PCSK9 loss induced apoptosis and inhibited G$_1$-S cell cycle progression (Fig. 2F, G). Western blot showed that PCSK9 depletion increased the expression of p27$^{Kip1}$, while reducing c-Myc, CDK4 and Cyclin

D1 (Fig. 2H). PCSK9 knockout in SW1116 cells also suppressed cell invasion and migration (Fig. 2I, J). These data suggest that PCSK9 functions as an oncogenic factor in *APC/KRAS*-mutant CRC cells.

We next investigated the effect of PCSK9 overexpression on *KRAS*-mutant CRC cells. PCSK9 overexpression in DLD1 cells promoted cell growth (Fig. 2K), inhibited cell apoptosis (Fig. 2L) and induced cell cycle progression (Fig. 2M). Western blot demonstrated increased c-Myc, CDK4 and cyclin D1/D3, but reduced p27$^{Kip1}$ (Fig. 2N). PCSK9 overexpression also promoted wound closure (Fig. 2O). PCSK9 overexpression thus induces malignant phenotypes in DLD1 cells in vitro.

**PCSK9-driven de novo cholesterol pathway activation and GGPP biosynthesis is required for the proliferation of APC/KRAS-mutant CRC.** We next determined the effect of PCSK9 on cholesterol homeostasis in CRC cells. Consistent with the role of PCSK9 in down-regulating LDLR, PCSK9 depletion increased BODIPY-LDL uptake in 1CT-AK and SW1116 cells, respectively (Fig. S6). We next asked if PCSK9 might activate SREBP2 by relieving cholesterol-mediated feedback inhibition. Indeed, *SREBF2* and cholesterol biosynthesis genes mRNA were down-regulated by PCSK9 knockdown/knockout in 1CT-AK and SW1116 cells (Fig. 3A), while PCSK9 overexpression in DLD1 cells exerted opposite effects (Fig. 3B). No consistent effects were observed with respect to *ABCA1*, *ABCG5*, *ABCG8* and *NPCL11*, suggesting that these cholesterol transporters were not modulated by PCSK9 (Fig. S2). Western blot showed that PCSK9 knockdown/knockout suppressed protein levels of mature (m) SREBP2, HMGCR, and SQLE in 1CT-AK, LOVO, and SW1116 cells (Fig. 3C). PCSK9 overexpression in DLD1 cells induced the expression of these genes (Fig. 3C). Knockdown of SERBP2 abolished effect of PCSK9 overexpression on the induction of cholesterol biosynthesis genes (Fig. S7), suggesting that PCSK9 regulates de novo cholesterol biosynthesis via SREBP2. PCSK9 inhibitors, PF-0644846 and R-IMPP, dose-dependently suppressed protein expression of mature SREBP2 and HMGCR in *APC/KRAS*-mutant CRC cells (Fig. 3D). D$_2$O labeling and LC-MS/MS confirmed repressed deuterium labeling of intracellular cholesterol in 1CT-AK and SW1116 cells transfected with *PCSK9*-siRNA without affecting total cholesterol levels, implying decreased de novo cholesterol biosynthesis (Fig. 3E). Conversely, PCSK9 overexpression in DLD1 cells promoted deuterium labeling of cholesterol (Fig. 3E). Consistent with

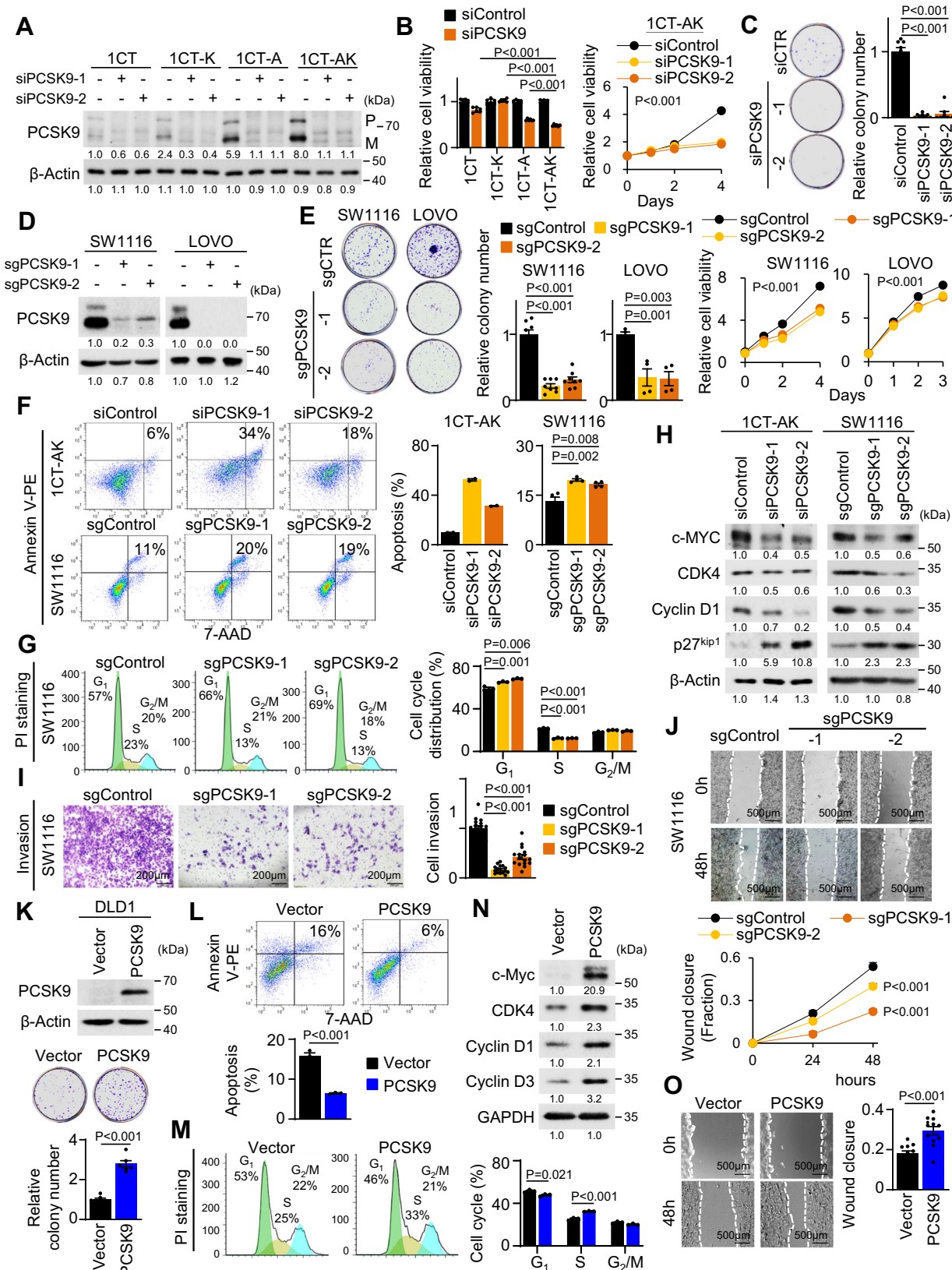

decreased metabolic flux via mevalonate pathway, PCSK9 knockdown in 1CT-AK cells reduced HMG-CoA, GPP, FPP and GGPP, whereas PCSK9 overexpression induced their levels in DLD1 cells (Fig. 3F). Moreover, PCSK9 also up-regulated GGPS1 expression via SREBP2 (Fig. S7). PCSK9 thus up-regulates de novo cholesterol pathway and GGPS1 to induce GGPP bio-synthesis in *KRAS*-mutant CRC.

To test our hypothesis that GGPP is the functional metabolite downstream of PCSK9 in *KRAS*-mutant CRC, we silenced geranylgeranyl pyrophosphate synthase (GGPS1), which catalyzes

**Fig. 2 PCSK9 promotes malignant phenotypes in *APC/KRAS*-mutant CRC. A** PCSK9 knockdown in 1CT isogenic cell lines. **B** PCSK9 knockdown suppressed cell proliferation ($n = 6$) and **C** colony formation ($n = 8$) in 1CT-AK cells. **D** PCSK9 knockout in SW1116 and LOVO cells. **E** PCSK9 knockout reduced cell growth ($n = 6$) and colony formation (SW1116: $n = 8$, 7 days; LOVO: $n = 4$, 14 days) in SW1116 and LOVO cells. **F** PCSK9 knockdown (72 h) or knockout induced cell apoptosis (1CT-AK: $n = 2$; SW1116: $n = 4$) and **G** inhibited G1-S cell cycle progression ($n = 3$). **H** Western blot revealed that PCSK9 knockdown/knockout induced the expression of cleaved PARP and p27$^{Kip1}$, while suppressing c-Myc, CDK4 and Cyclin D1. **I** PCSK9 knockout suppressed Matrigel invasion (72 h, 16 replicates in 4 independent experiments) and **J** wound healing closure (12 replicates in 4 independent experiments) in SW1116 cells. **K** PCSK9 overexpression in DLD1 cells increased cell proliferation ($n = 6$, 7 days), **L** inhibited apoptosis ($n = 3$) and **M** promoted G1-S progression ($n = 3$). **N** PCSK9 increased CDK4, Cyclin D1, and Cyclin D3, but suppressed p27$^{Kip1}$ in DLD1 cells. **O** PCSK9 overexpression promoted cell migration in DLD1 cells (48 h, 12 replicates in 4 independent experiments). Data shown are means of biological replicates; ± SEM (**B**, **C**, **E**–**G**, **I**–**M**, **O**). Two-tailed Student's *t*-test (**B**, **C**, **E**–**G**, **I**, **K**–**M**, **O**). Two-tailed two-way ANOVA (growth curves in **B**, **E**, **J**). Source data are provided as a Source Data file.

the biosynthesis of GGPP, in PCSK9-overexpressing DLD1 cells. GGPS1 knockdown abolished the growth promoting effect of PCSK9 (Fig. 3G), implying that GGPP functions downstream of PCSK9 to promote cell growth. GGPS1 knockdown in SW1116 and LOVO cells with high PCSK9 expression also impaired cell proliferation in vitro and SW1116 xenografts growth in vivo (Fig. S8). In addition, we incubated the sgControl and PCSK9 knockout cells with cholesterol or its intermediary metabolites (GGPP, FPP, MVA: mevalonate, SQE: squalene, and CHO: cholesterol) at 10 µM. Only GGPP rescued cell viability of PCSK9 knockout LOVO cells, which was validated in SW1116 cells (Fig. 3H), highlighting a role of GGPP in contributing to the oncogenic effect of PCSK9.

**PCSK9 promotes KRAS/MEK/ERK signaling cascade.** GGPP is involved in protein prenylation, an obligatory modification for small GTP-binding protein (GTPases) essential for membrane localization and activation to promote downstream signaling[15]. To identify pathways activated by PCSK9-GGPP, we employed phospho-kinase antibody arrays. Among 24 kinases, p-ERK1/2 was the most significantly down-regulated kinase following PCSK9 depletion (Fig. 4A). As KRAS is a key upstream GTPase that activates p-MEK/p-ERK, we analyzed KRAS activation by the c-RAF-RBD-GST pulldown assay (Fig. 4B). 1CT-AK cells exhibited higher expression of active KRAS compared to 1CT-A cells (Fig. S9). Interestingly,

knockdown of PCSK9 decreased active KRAS levels in 1CT-AK cells, but had no corresponding effects on 1CT-A cells (Fig. 4B and Fig. S9). PCSK9 knockdown also repressed active KRAS in SW1116 cells (Fig. 4B). PCSK9 loss was associated with decreased expression of membrane-localized KRAS in 1CT-AK and SW1116 cells (Fig. 4C). Conversely, PCSK9 overexpression in DLD1 cells increased activated KRAS and promoted its membrane localization (Fig. 4D). In addition, PF-0644846 or R-IMPP inhibited active KRAS expression in SW1116 cells (Fig. 4E).

Consequently, depletion of PCSK9 suppressed p-MEK, p-ERK, p-AKT, and downstream target p-p90S6K in 1CT-AK and SW1116 cells (Fig. 4F and Fig. S9), while they were all up-regulated in PCSK9-overexpressing DLD1 cells (Fig. 4G). Consistently, PCSK9 inhibitors suppressed p-ERK and cyclin D3 in KRAS-mutant CRC cells (Fig. 4H). The activation of KRAS/MEK/ERK depends on the PCSK9-GGPP axis. GGPS1 knockdown abrogated PCSK9 expression-induced p-MEK/p-ERK expression in DLD1 cells (Fig. 4I), whereas GGPP supplementation restored p-MEK/p-ERK in LOVO cells with PCSK9 knockout (Fig. 4J) or treated with PCSK9 inhibitors (Fig. S10). Blockade of MEK1/2 abrogated growth promoting effect of PCSK9 overexpression in DLD1 cells (Fig. S11), consistent with its role in promoting cell growth downstream of the PCSK9-GGPP axis. PCSK9 therefore promotes cell growth via GGPP-dependent activation of KRAS/MEK/ERK in APC/KRAS-mutant CRC (Fig. 4K).

**PCSK9 knockout inhibits tumorigenicity of *APC/KRAS*-mutant CRC in vivo.** In light of our in vitro findings, we next investigated the effect of PCSK9 in vivo. PCSK9 knockout suppressed the growth of SW1116 ($P < 0.01$) and LOVO cells ($P < 0.05$) subcutaneous xenografts by over 50%, both in terms of tumor volume and tumor weight (Fig. 5A and Fig. S12). PCSK9 knockout inhibited cell proliferation and induced apoptosis in vivo, as evidenced by Ki-67 and TUNEL staining, respectively (Fig. 5B). On the other hand, PCSK9 overexpression accelerated the growth of DLD1 xenografts in nude mice ($P < 0.01$) (Fig. 5C) by inducing cell proliferation (Ki-67) and inhibiting apoptosis (TUNEL) (Fig. 5D). Western blot demonstrated that PCSK9 knockout inhibited protein expression of p-ERK, Cyclin D3 and CDK4 in SW1116 xenografts (Fig. 5E), while PCSK9 over-expression increased p-ERK, Cyclin D3, and CDK4 expression in DLD1 xenografts (Fig. 5E). Metabolomic analysis demonstrated that PCSK9-overexpressing tumors have elevated GGPP ($P < 0.01$); whereas PCSK9 knockout suppressed GGPP in SW1116 xenografts ($P < 0.05$) (Fig. 5F), confirming that PCSK9-GGPP axis is operative in vivo. We also tested the effect of PCSK9 loss ex vivo in primary CRC organoids isolated from $Apc^{Min/+}Kras^{G12D/+}$ Villin-Cre mice. shPcsk9 inhibited proliferation of $Apc^{Min/+}Kras^{G12D/+}$ Villin-Cre tumor organoids in Matrigel (Fig. S13). Our data collectively indicate that PCSK9 is an oncogenic factor in vivo in APC/KRAS mutant CRC.

**PCSK9 inhibitors suppressed *APC/KRAS*-mutant CRC growth in vitro and in vivo.** PCSK9 inhibitors consist of two classes: anti-PCSK9 antibodies and PCSK9 translation inhibitors. We first evaluated the effect of Evolocumab (anti-PCSK9 antibody). DLD1 and 1CT-AK cells showed repressed growth in 10 and 40 µg/ml Evolocumab, while no effect was observed in 1CT cells (Fig. 5G). PCSK9 translation inhibitors R-IMPP and PF-0644846 were more cytotoxic towards 1CT-AK and DLD1 cells compared to 1CT cells (Fig. 5H). Dose-response study (72h-IC$_{50}$) validated that *KRAS*-mutant CRC cells were more sensitive to R-IMPP and PF-0644846 compared to 1CT and NCM460 normal colonic cell lines (Fig. 5H and Fig. S14). R-IMPP and PF-0644846 also had no effect on apoptosis in NCM460 cells (Fig. S14). PCSK9 inhibitors exerted significant inhibitory effects in human *APC/KRAS*-mutant CRC primary organoids (Fig. 5I), and inhibited cholesterol biosynthesis pathway, as determined by qPCR and western blot (Fig. 5J). On the contrary, PCSK9 inhibitors did not have consistent effect on *ABCA1* in CRC organoids (Fig. S2). PCSK9 inhibitors had no effect on *KRAS*-wildtype CRC organoids (Fig. S15), inferring its selectivity for *APC/KRAS*-mutant CRC.

We next investigated the anticancer efficacy of PCSK9 inhibitors in vivo. Mice harboring SW1116 xenografts (~150 mm³) were randomized, and treated i.p. with vehicle, R-IMPP (100 mg/kg) or PF-0644846 (50 mg/kg). R-IMPP ($P < 0.01$) and PF-0644846 ($P < 0.05$) suppressed growth of SW1116 xenografts, with decreased tumor volume and weight (Fig. 6A). No significant decrease in body weight was observed in treatment groups. Administration of

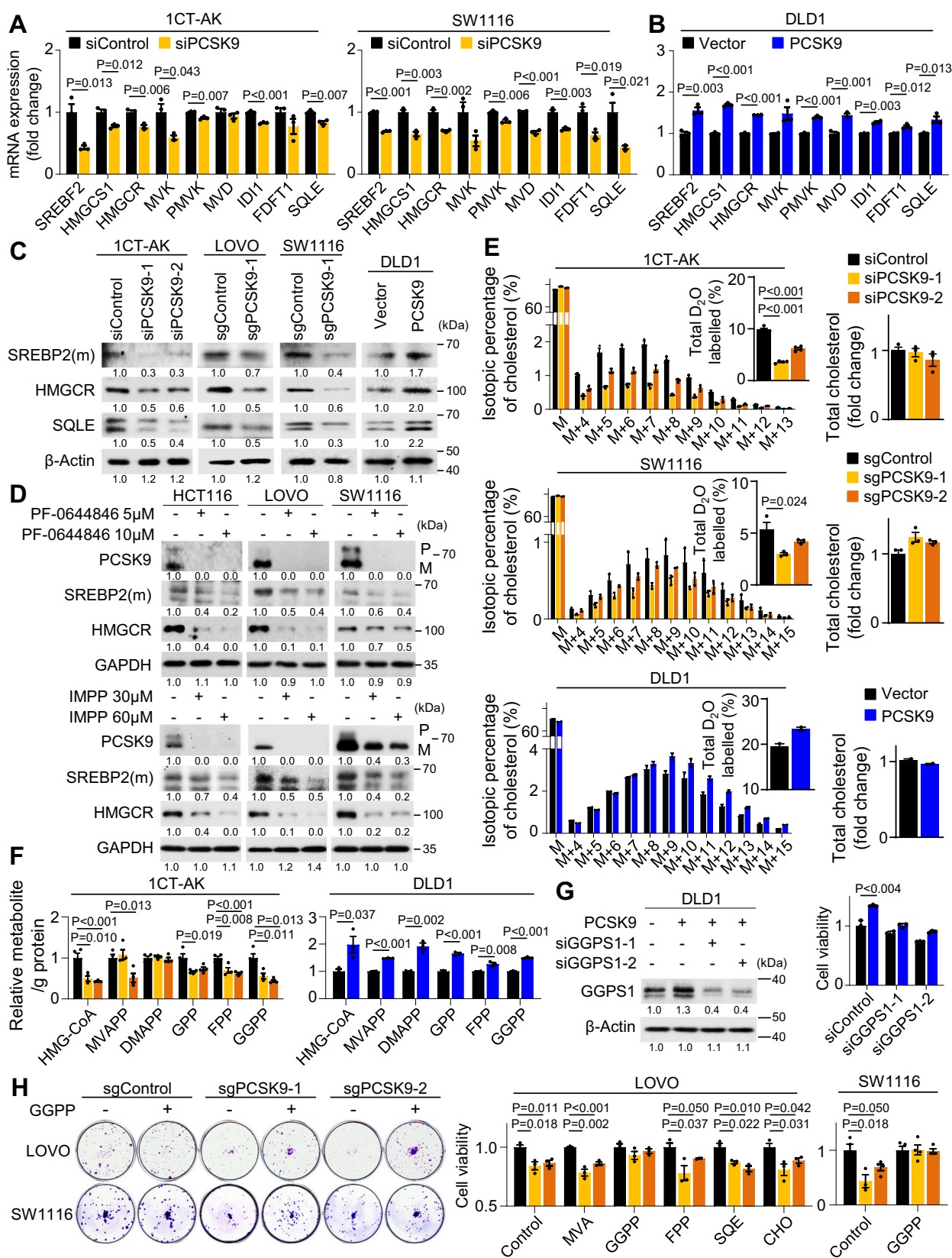

R-IMPP to $Apc^{Min/+}Kras^{G12D/+}Villin\text{-}Cre$ (50 mg/kg, i.p., 3 times/ week) also significantly suppressed tumor number ($P < 0.01$) and load ($P < 0.01$) (Fig. S16), validating its efficacy in spontaneous *APC/KRAS*-mutant CRC. Together, PCSK9 inhibitors suppressed *KRAS*-mutant CRC growth in vivo at a dose that is apparently safe.

**PCSK9 inhibitors synergize with HMGCR inhibitors to suppress *APC/KRAS*-mutant CRC.** HMGCR inhibitors suppress cancer cell growth; however, drug resistance arises from cholesterol depletion and subsequent up-regulation of SREBP2 and HMGCR. As PCSK9 inhibition reduced SREBP2 and HMGCR

**Fig. 3 PCSK9 activates de novo cholesterol biosynthesis and GGPP accumulation to promote cell growth. A** mRNA expression of *SREBF2* and cholesterol biosynthesis genes was decreased in PCSK9 knockdown 1CT-AK cells and PCSK9 knockout SW1116 cells ($n = 3$). **B** Ectopic expression of PCSK9 in DLD1 cells promoted *SREBP2* and cholesterol biosynthesis genes mRNA expression ($n = 3$). **C** Western blot showed that PCSK9 knockdown or knockout reduced protein expression of SERBP2 (mature form), HMGCR, and SQLE in 1CT-AK, LOVO, and SW1116 cells; PCSK9 overexpression in DLD1 cells exerted an opposite effect. **D** R-IMPP and PF-0644846, inhibitors of PCSK9, suppressed expression of SREBP2, HMGCR, and SQLE in *APC/KRAS*-mutant CRC cell lines. **E** $D_2O$-labeling (48 h) and LC-MS analysis of intracellular cholesterol demonstrated that PCSK9 depletion inhibited de novo cholesterol biosynthesis in 1CT-AK ($n = 4$) and SW1116 cells ($n = 3$), while PCSK9 overexpression promoted de novo cholesterol biosynthesis in DLD1 cells ($n = 2$). **F** PCSK9 depletion reduced HMG-CoA, GPP, FPP and GGPP levels in 1CT-AK cells ($n = 4$), while PCSK9 overexpression increased GPP, FPP and GGPP in DLD1 cells ($n = 4$). **G** GGPS1 knockdown ($n = 3$, 72 h) abrogated growth inducing effect of PCSK9 overexpression in DLD1 cells. **H** Effect of cholesterol and intermediary metabolites in rescuing the proliferation of PCSK9 knockout SW1116 ($n = 4$) and LOVO ($n = 3$) cells. Data shown are means of biological replicates; ± SEM (**A**, **B**, **E**–**H**). Two-tailed Student's *t*-test (**A**, **B**, **E**–**H**). Source data are provided as a Source Data file.

expression, we predicted that co-inhibition of HMGCR and PCSK9 would synergistically or additively suppress cell growth. We treated PCSK9 knockdown 1CT-AK cells with simvastatin and found that simvastatin 72h-IC$_{50}$ values were reduced by >6-fold in PCSK9 knockdown cells (Fig. 6B). Simvastatin (10 μM) plus si*PCSK9* also synergistically triggered apoptosis (Fig. 6B). We next evaluated the combination of PCSK9 inhibitors with sim-vastatin in *KRAS*-mutant CRC cell lines. Evolocumab in combination with simvastatin increased apoptosis in DLD1 and HCT116 cells over individual inhibitors (Fig. 6C). To quantitatively determine if the effects were synergistic, we performed isobologram analysis of PCSK9 inhibitors-simvastatin drug combinations in DLD1, HCT116, and SW1116 cells (48 h-IC$_{50}$). R-IMPP or PF-0644846 plus simvastatin were determined to synergistically inhibit cell growth, as exemplified by combinations below the additivity line in the isobolograms (Fig. 6D). Finally, we assessed the combination of R-IMPP (50 mg/kg) plus simvastatin (30 mg/kg) in vivo using SW1116 xenografts in nude mice (Fig. 6E). R-IMPP plus simvastatin triggered tumor regression compared to control or single drug treatments ($P < 0.01$), and reduced tumor weight ($P < 0.05$) (Fig. 6E). PCSK9 protein and GGPP were down-regulated by R-IMPP plus simvastatin, supporting a role of GGPP inhibition in suppressing tumor growth (Fig. 6E). Collectively, PCSK9 inhibitors in conjunction with statins are synergistic in inhibiting *APC/KRAS*-mutant CRC growth.

**PCSK9 is overexpressed in primary CRC tumors and is associated with poor prognosis in CRC harboring mutant *APC/KRAS*.** To evaluate the clinical significance of PCSK9 in CRC, we analyzed its expression in a series of CRC and adjacent normal tissues. The up-regulation of *PCSK9* mRNA in CRC was independently demonstrated in paired tumor and adjacent normal tissues in Hong Kong ($N = 150$; $P < 0.0001$) and TCGA COADREAD cohorts ($N = 50$; $P < 0.0001$) (Fig. 6F), and in unpaired CRC ($N = 624$) and adjacent normal tissues ($N = 51$) from TCGA ($P < 0.0001$) (Fig. 6F). PCSK9 protein levels was also up-regulated in primary CRC compared to adjacent normal tissues as determined by western blot (Fig. 6G).

To evaluate the prognostic significance of PCSK9 in CRC patient survival and its relationship with KRAS mutation status, we examined PCSK9 expression using a tissue microarray consisting of 137 CRC patients with known *APC* and *KRAS* mutation status (Table S2). Kaplan-Meier curve revealed that PCSK9 high protein expression was associated with poor survival in specifically in *APC/KRAS*-mutant CRC ($P < 0.005$), but not in *APC*-mutant only CRC, or the overall CRC cohort (Fig. 6H). We then examined the prognostic significance of PCSK9 in TCGA COADREAD cohort ($N = 368$) (Table S3). Kaplan-Meier curves showed that high *PCSK9* mRNA expression correlated with poor survival only in *APC/KRAS*-mutant CRC patients ($P < 0.05$), but not in *APC*-mutant only CRC or overall cohort (Fig. 6I).

Multivariate COX proportional hazards regression analysis showed that PCSK9 protein is an independent prognostic marker for *APC/KRAS*-mutant CRC in Hong Kong ($P < 0.005$) and TCGA ($P < 0.05$) cohorts (Fig. 6J and Tables S4 and S5). PCSK9 predicts poor prognosis in CRC patients harboring *APC/KRAS* mutations, in line with the oncogenic function of PCSK9 in *APC/KRAS*-mutant CRC.

Finally, we sought to determine the association of PCSK9 to cholesterol biosynthesis, GGPS1-GGPP and MEK-ERK signaling in human CRC. As shown in Fig. S17, *PCSK9* expression is positively correlated with cholesterol biosynthesis (*SREBP2*, *HMGCR*, *FDFT1*, *SQLE*), *GGPS1*, and MEK-ERK (*CCND1*, *CCNE1*) in human CRC patients. Moreover, GGPS1 protein predicts the poor survival in *APC/KRAS*-mutant CRC ($P < 0.05$) but not in *APC*-mutant CRC in Hong Kong cohort (Fig. S17). Our results support our notion that PCSK9 induces cholesterol biosynthesis, GGPPS1 and MEK-ERK signaling in human CRC.

## Discussion

Mutant *KRAS* poses a major challenge for the management of CRC patients. Our investigation demonstrated that sequential mutations in *APC* and *KRAS* rewires cholesterol homeostasis in CRC cells, which mediates a positive feedforward cycle to induce KRAS activation and its downstream signaling involved in cell growth. Mechanistically, PCSK9 plays a pivotal role in this process whereby it represses exogenous cholesterol uptake but induced de novo cholesterol biosynthesis, culminating in the accumulation of GGPP, an intermediate that can activate KRAS. Genetic and pharmacological blockade of PCSK9 suppressed *APC/KRAS*-mutant CRC proliferation. Our data illustrate the importance of deregulated cholesterol metabolism in *APC/KRAS*-mutant CRC and inform future therapeutic strategies.

Cellular cholesterol homeostasis is maintained via the import of circulating lipoproteins or de novo biosynthesis via the mevalonate pathway. Using isogenic cell lines expressing wild-type or mutant *KRAS*, we demonstrated that mutant *KRAS* in conjunction with alterations in *APC* altered the mode of cholesterol acquisition in human colonic cells, while having no discernible impact on the total cholesterol levels. Uptake assays with fluorescent-labelled LDL revealed the dramatic reduction of LDL uptake in colon cells expressing sh*APC* plus *KRAS*$^{G12V}$. In contrast, stable isotope labeling with $D_2O$ as the tracer showed that normal colonic cells exhibit minimal de novo cholesterol biosynthesis, which was significantly promoted upon co-expression of sh*APC* and *KRAS*$^{G12V}$. In line with this observation, viability of normal colon cells is highly dependent on exogenous supply of lipoproteins, whereas sh*APC* plus *KRAS*$^{G12V}$ confers a capacity to proliferate relying on de novo biosynthesis. Indeed, most normal cells utilize circulating lipids (e.g., LDL, VLDL) to satisfy their requirement for lipids and cholesterol, and enhanced cholesterol biosynthesis is a hallmark of cancer cells[16–20]. While some cancers such as breast cancers and glioblastoma are known to induce

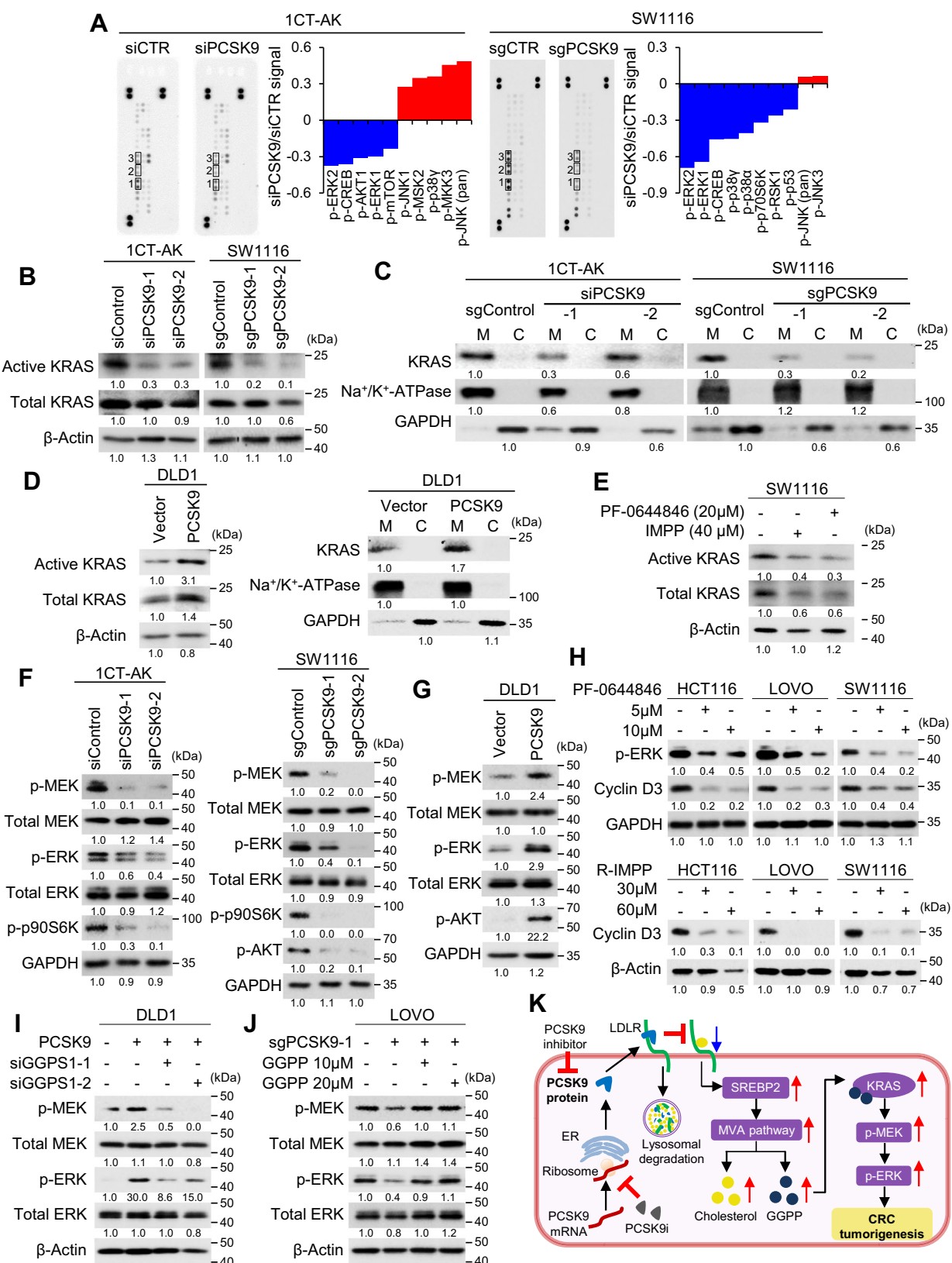

cholesterol uptake via LDLR[21,22], de novo cholesterol biosynthesis is often up-regulated in CRC[16]. Mutant *KRAS* and *APC*-loss thus induces a pivotal shift in the source of intracellular cholesterol in colonic cells.

Enhanced cholesterol biosynthesis in *APC/KRAS*-mutant CRC is mediated via concerted action of PCSK9 and cholesterol

biosynthesis genes. PCSK9 binds to lipoprotein receptors such as LDLR and promotes LDLR intracellular degradation, thereby repressing uptake of exogenous lipids[23]. We observed remarkable up-regulation of PCSK9 in isogenic cells with mutant *APC/KRAS*, in transgenic mice with colon-specific sh*Apc* and *Kras*[G12D] expression and in human CRC. Increased PCSK9 is accompanied

**Fig. 4 PCSK9 promotes KRAS/MEK/ERK signaling cascade via GGPP. A** Phospho-MAPK antibody array assay of 1CT-AK and SW1116 cells with PCSK9 knockdown or knockout, respectively. p-ERK1/2 and p-CREB were consistently down-regulated. **B** Active KRAS pulldown assays revealed that PCSK9 depletion suppressed KRAS activation. **C** Loss of PCSK9 decreased the membrane localization of KRAS. **D** PCSK9 overexpression in DLD1 cells increased active KRAS expression and its membrane localization. **E** PCSK9 inhibitors, R-IMPP and PF-0644846, suppressed the activation of KRAS in 1CT-AK and SW1116 cells. **F** PCSK9 knockdown/knockout in 1CT-AK and SW1116 cells inhibited phosphorylation of MEK, ERK, and p09S6K. **G** PCSK9 overexpression induced p-MEK and p-ERK in DLD1 cells. **H** R-IMPP and PF-0644846 inhibited p-ERK and expression of its downstream target Cyclin D3 in *APC/KRAS*-mutant CRC cell lines. **I** GGPS1 knockdown abrogated the induction of p-MEK and p-ERK by PCSK9 overexpression in DLD1 cells. **J** Supplementation of GGPP rescued p-MEK and p-ERK expression in PCSK9 knockout LOVO cells. **K** Schematic diagram summarizing the molecular mechanism of PCSK9 in APC/KRAS-mutant CRC. Source data are provided as a Source Data file.

by reduced LDLR and LDL uptake, which in turn, up-regulated expression of mature SREBP2, HMGCR and SQLE, the rate limiting enzymes of cholesterol biosynthesis. Corroborating these findings, $D_2O$-labeling confirmed that PCSK9 positively regulates cholesterol biosynthesis in *KRAS*-mutant CRC cells. We further identified that PCSK9 up-regulation is driven by β-catenin, representing a point of convergence between the WNT and KRAS signaling cascade. PCSK9 expression is also positively regulated by SREBP2 and E2F1[23,24], while being repressed by FOXO3[25]. Since PCSK9 up-regulated SREBP2 in CRC cells, β-catenin-driven PCSK9 might drive a positive feedback loop involving PCSK9 and SREBP2. Our work unravels the underlying role of PCSK9 up-regulation on altered cholesterol homeostasis in *APC/KRAS*-mutant CRC cells.

Cholesterol biosynthesis imparts a high energetic cost for proliferating cells, and it is unclear why cancer cells rely on this ATP-consuming process. Analysis of intermediary metabolites showed that increased cholesterol biosynthesis drives GGPP accumulation in *KRAS*-mutant CRC cells and in tumors from $Apc^{Min/+}Kras^{G12D/+}$ *Villin-Cre* mice. Several lines of evidence imply GGPP as the functional downstream metabolite of PCSK9 in *APC/KRAS*-mutant CRC. First, PCSK9 positively regulates GGPP biosynthesis. Moreover, knockdown of GGPS1 abrogated the growth-promoting effect of PCSK9. Finally, the supplementation of GGPP, but not other intermediary metabolites, restored viability of PCSK9 depleted cells. Hence, up-regulation of PCSK9 and engagement of the energy-consuming mevalonate pathway is offset by the biosynthesis of GGPP, an oncometabolite for *APC/KRAS*-mutant CRC.

GGPP is an isoprenoid involved in the prenylation of small GTPases, such as KRAS[26], thereby enhancing their membrane localization and activation. Consistent with this notion, disruption of PCSK9-GGPP axis by knockout of PCSK9 in *APC/KRAS*-mutant CRC cells were found to reduce active KRAS and repress its membrane localization. This resulted in the inhibition of MEK/ERK signaling and decreased expression of downstream genes (Cyclin D1/D3 and CDK4) involved in cell proliferation, an effect reversed by GGPP. Conversely, overexpression of PCSK9 activated KRAS/MEK/ERK, which was abolished by knockdown of GGPS1. These data indicate that the PCSK9-GGPP axis is directly involved in the activation of KRAS/MEK/ERK oncogenic signaling in *APC/KRAS*-mutant CRC.

PCSK9 is a therapeutic target in hypercholesterolemia[27], but is also emerging as a potential target in cancer[28,29]. PCSK9 inhibitors have been approved by FDA and more are under development. Evolocumab, R-IMPP, and PF-0644846 inhibited the growth of APC/KRAS-mutant CRC cells and CRC organoids. Xenograft models confirmed the efficacy of R-IMPP and PF-0644846 in vivo without overt adverse effects. Based on the premise that HMGCR inhibitors suppress GGPP levels, we devised the combination of PCSK9 inhibitors and simvastatin. si*PCSK9* or PCSK9 inhibitors synergized with simvastatin to inhibit cell growth and induce apoptosis in vitro, and demonstrated a synergistic effect in

SW1116 xenografts. Our work, together with that of others[16,30], highlights mevalonate pathway as an emerging cancer drug target for. Mevalonate pathway blockers such as statins and Tasin-1 have been shown to suppress glycolysis[31], pyrimidine bioynthesis[30], and induce cell death in cancer cells[16]. By combining statin and PCSK9 inhibitors, a more effective blockade of mevalonate pathway could be achieved, thereby further improving therapeutic efficacy.

In human CRC, *APC/KRAS* double mutants are present in 34.1% and 36.7% of CRC, according to TCGA cohort and our TMA cohort. Here, we revealed that PCSK9 expression is an independent prognostic factor associated with poor survival in *APC/KRAS* mutant CRC, but not in single *APC* mutant counterparts. Hence, PCSK9 is a therapeutic target and prognostic biomarker for this CRC subtype.

In conclusion, our findings highlight an important role of PCSK9 in inducing GGPP-dependent activation of KRAS/MEK/ERK signaling cascades to promote cell growth. Moreover, PCSK9 can be targeted in combination with statins to suppress *APC/KRAS*-mutant CRC in vitro and in vivo. Thus, our work defines PCSK9 as an oncogenic factor and a potential therapeutic target in *APC/KRAS*-mutant CRC.

## Methods

**Study approvals**. For human samples, informed consent was obtained for all patients and this study was approved by ethics committee of the Chinese University of Hong Kong and the Beijing University Cancer Hospital. All animal work is approved by the Animal Experimentation Ethics Committee of the Chinese University of Hong Kong.

**Human samples**. Three cohorts of CRC patients were included in our analysis. The first CRC cohort was obtained from the Beijing University Cancer Hospital and consists of 150 CRC with surgically excised CRC tissues and surrounding adjacent normal tissues that were used for PCSK9 mRNA expression by qPCR. Specimens were snap frozen in liquid nitrogen and stored at −80 °C until analysis. The second cohort was a tissue microarray (TMA) generated from formalin-fixed, paraffin-embedded CRC tissues of 180 patients collected at the Prince of Wales Hospital, Hong Kong (9–10), with median follow-up time of 61.1 months. Immunohistochemistry was performed using anti-PCSK9 (MA542843; Thermo-Fisher). Scoring was performed by a pathologist blinded to the nature of the samples (Dr. H.K. Cheung, CUHK). All the subjects provided informed consent for the study specimens. TCGA COADREAD cohort consists of 50 adjacent normal and 624 CRC samples, among which 478 CRC samples with *KRAS* mutation and survival status were used for analysis. Informed consent was obtained for all patients and this study was approved by ethics committee of the Chinese University of Hong Kong and the Beijing University Cancer Hospital.

**Cell culture**. DLD1, HCT116, LOVO, and SW1116 cell lines were obtained from the American Type Culture Collection (Rockville, MD). DKS8 cells were from Prof. Senji Shirasawa, Fukuoka University. These cell were cultured in DMEM supplemented with 10% FBS and penicillin-streptomycin. 1CT normal colonic cells and 1CT cells expressing sh*APC* (1CT-A) was obtained from Prof. Jerry Shay (UT Southwestern Medical Center, TX)[12]. Both of the cell lines were transduced with lentiviral-*KRAS*$^{G12V}$ (Addgene: 35634) to provide 1CT-*KRAS*$^{G12V}$ (1CT-K) and 1CT-sh*APC*-*KRAS*$^{G12V}$ cells (1CT-AK).

**Primary CRC organoid culture**. Human primary *KRAS*-mutant CRC organoids (CRC-74 and CRC-112) was obtained from the Princess Margaret Living Biobank,

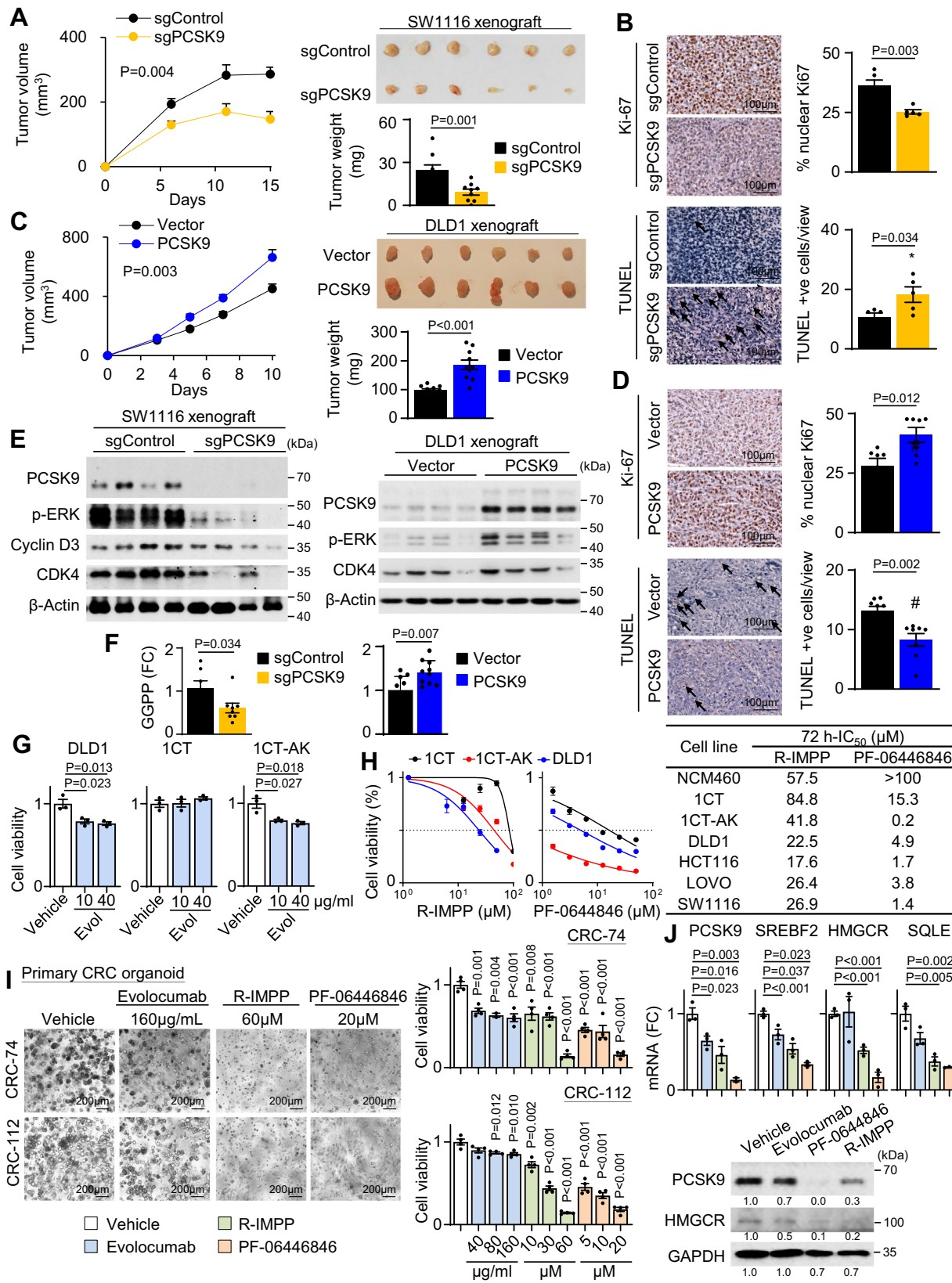

University of Toronto, Canada. Informed consent was obtained from patients for providing the study specimens. *APC* and *KRAS* mutation status were confirmed by sequencing of primary tumor tissues. They were cultured in 3D Matrigel in DMEM/F12 medium with 10 mM HEPES, 2 mM Glutamax, B27, 50 ng/ml mouse EGF, 10 nM [Leu15]-Gastrin I, 100 ng/ml Noggin, 0.5 μM A83-01, 1.0 mM N-acetylcysteine and penicillin-streptomycin. Mouse *KRAS*-mutant CRC organoid was isolated from *Apc^Min/+Kras^G13D/+villin-Cre* mice tumor[9] and then transduced

with *Pcsk9* shRNA-GFP (TL505587, Origene, Rockville, MD). Knockdown organoids were selected by cell sorter based on GFP fluorescence.

**Knockdown, knockout and overexpression in CRC cell lines**. Knockdown was performed with silencer select siRNA (PCSK9: s48695 and s48696; SREBP2: s27 and s29; GGPS1 s18108 and s18109, ThermoFisher) and transfected using

**Fig. 5 PCSK9 promotes *APC/KRAS*-mutant CRC growth in vivo and growth inhibitory effect of PCSK9 inhibitors. A** PCSK9 knockout suppressed growth of subcutaneous xenograft model of SW1116 cells in NOD/SCID mice ($n = 1$ experiment, $n = 9$ mice per group). **B** PCSK9 knockout inhibited cell growth as indicated by Ki-67 staining ($n = 5$); TUNEL showed that PCSK9 knockout induced apoptosis in SW1116 xenografts ($n = 5$). **C** PCSK9 overexpression in DLD1 cells induced xenograft growth in nude mice ($n = 1$ experiment, $n = 10$ mice per group). **D** PCSK9 overexpression in DLD1 cells induced cell growth and inhibited apoptosis, as evidenced by Ki-67 ($n = 7$) and TUNEL ($n = 8$) assays, respectively. **E** Western blot of SW1116 xenografts revealed that PCSK9 knockout suppressed p-ERK, Cyclin D1, and CDK4. Western blot of DLD1 xenografts demonstrated increased expression of p-ERK, Cyclin D1, and CDK4. **F** GGPP levels in SW116 xenografts with PCSK9 knockout ($n = 8$) and DLD1 xenografts with PCSK9 overexpression ($n = 10$). **G** Effect of Evolocumab on 1CT, 1CT-AK, and DLD1 cells ($n = 3$). **H** Effect of R-IMPP and PF-0644846 on 1CT, 1CT-AK, and DLD1 cells ($n = 3$). 72 h-IC$_{50}$ values of R-IMPP and PF-0644846 in a panel of *APC/KRAS* mutant CRC, 1CT isogenic cells and NCM460 cells. **I** Effect of Evolocumab, R-IMPP, and PF-0644846 on cell viability of primary organoids from human *APC/ KRAS*-mutant CRC after 5–7 days of treatment ($n = 4$). **J** Effect of Evolocumab, R-IMPP and PF-0644846 on cholesterol gene expression in *KRAS*-mutant CRC organoids ($n = 3$) after 72 h of drug treatment. Data shown are means of biological replicates; ± SEM (**A**–**D**, **F**, **G**–**J**). Two-tailed Student's *t*-test (**A**–**D**, **F**, **G**, **I**, **J**). Two-tailed two-way ANOVA for growth curves (**a**, **c**). Source data are provided as a Source Data file.

---

Lipofectamine RNAiMAX reagent in Opti-MEM. Control-siRNA was from Ribobio (Guangzhou). For PCSK9 knockout, we cloned following sgRNA sequences: sg1: 5'-CGCAAGGCTCAAGGCGCCGC-3'; and sg2: 5'-GGACGAGGACGGCGACTACG-3' into LentiCRISRPv2 vector (Addgene, #52961). 293 T cells were used to package the lentivirus-sgRNA together with packaging vector. Transduced cells were selected with 2 µg/ml puromycin and single clones were validated by western blot. For overexpression, DLD1 cells were transfected with pCMV3-PCSK9 (Sinobiological, lnc) or empty vector, followed by selection with 200 µg/ml hygromycin for 14 days to obtain stable cell lines.

**Metabolite assessment**. D$_2$O labelling was performed with DMEM prepared in D$_2$O:H$_2$O (50:50, v/v), and incubated for the indicated duration. For cholesterol analysis, cells or tissues were washed with PBS twice, and homogenized in 400 µl of ice-cold methanol by using a Polytron PT2100 homogenizer. 400 µl of chloroform was then added for 1 h, followed by 300 µl of water. After equilibrating on ice 10 min, homogenate was centrifuged at 14000 g for 10 min at 4 ºC. Lower organic layer was reconstituted in 100 µl of 90% methanol, 10 mM ammonium acetate with $^{13}$C$_2$-cholesterol (50 ppm) as internal standard. LC-MS analysis was carried on a UPLC-Q Exactive MS equipped with an electrospray ion source (ThermoFisher Scientific). A Waters BEH C18 analytical column (1.7 µm, 2.1 mm x 100 mm) was employed for the separation, being kept at 55 ºC during the analysis. MS analysis was operated in positive ionization mode, with full scan mode from m/z 368 to 400 with resolution of 70000. Quantification of individual cholesterol isotopomers were performed with a 4ppm window centered on the theoretical m/z. Metabolite signal were normalized to internal standard and protein content for cell pellet, or weight for tissue samples.

For MVA metabolite analysis, cells or tissues were washed with PBS and homogenized in 500 µl of a mixture of methanol, acetonitrile and water (4:4:2) with 1% ammonium hydroxide and the internal standard 4-Chloro-phenylalanine (0.5ppm) by using a Polytron PT2100 homogenizer. The extract went through 3 freeze-thaw cycles under liquid N$_2$, centrifuged (21500 g, 15 min, 4 ºC), and the supernatant was dried and reconstituted in 100 µl of 50% methanol with 1% ammonium hydroxide. MVA analysis was carried on a Thermo Scientific UPLC system coupled to a TSQ Quantiva™ Triple Quadrupole MS equipped with an ESI source. The chromatographic separation was achieved on a Waters BEH C8 column (2.1 mm × 100 mm, 1.7 µm) in 15 min, kept at 35 ºC. The mobile phases were water with 10 mM ammonium acetate and 0.1% (v/v) ammonium hydroxide (A) and acetonitrile (B). Mass spectrometry was performed in negative ion multiple reaction monitoring mode. Metabolite signal were normalized to internal standard and protein content for cell pellet, or weight for tissue samples.

**Xenograft models**. Cells ($2-5 \times 10^6$ cell in 0.1 ml PBS) were injected subcutaneously into left and right dorsal flanks of six-weeks-old female nude mice. Tumor size was measured at intervals with a caliper. For drug treatment, mice ($N = 5$ per group) were randomized when tumor sizes reached 100–150 mm$^3$. For single PCSK9 inhibitor study, mice were grouped into 1) vehicle; 2) R-IMPP (100 mg/kg); and 3) PF-06446846 (50 mg/kg) and treated intraperitoneally every 2 days. For combination study, there are 4 groups: 1) vehicle; 2) simvastatin (30 mg/kg); 3) R-IMPP (50 mg/kg), 4) simvastatin+R-IMPP, twice a week i.p. All the drugs were dissolved in PBS, and vehicle groups were given PBS only injection. Tumor volume was calculated using the following formula: tumor volume = [length × width × (length + width/2) × 0.56]. Maximal tumor size permitted was >1.25 cm in any dimension, and the limit was not exceeded. All animal work is approved by the Animal Experimentation Ethics Committee of the Chinese University of Hong Kong.

**Apc$^{min/+}$Kras$^{G12D}$Villin-Cre mice model**. *Apc$^{min/+}$Kras$^{G12D}$Villin-Cre* mice model was provided by Nanjing Biomedical Research Institute, Nanjing University. For metabolomic assays, 7-weeks-old male *Apc$^{min/+}$Kras$^{G12D}$Villin-Cre* male mice

were harvested, and paired CRC and adjacent normal tissues were snap frozen and stored at −80 ºC. For treatment studies, *Apc$^{min/+}$Kras$^{G12D}$Villin-Cre* male mice were treated with vehicle (PBS) or R-IMPP (50 mg/kg in PBS, i.p., 3 times a week) from week 4 to week 7. Mice were sacrificed on week 7 and tumor number and size were measured. Tumor load was calculated from the sum of tumor volume in each mouse using the following formula: tumor volume = [length × width × (length + width/2) × 0.56]. All animal work is approved by the Animal Experimentation Ethics Committee of the Chinese University of Hong Kong.

**Western blot**. Whole cell lysates were prepared from cell or tissues with Cytobuster protein extraction reagent supplemented with cOmplete Mini protease inhibitor cocktail (Roche) and PhosSTOP (Roche). Membrane proteins were harvested using Mem-PER Plus Membrane Protein Extraction Kit (ThermoFisher). Protein lysates (10–30 µg) were separated by SDS-PAGE, and transferred to PVDF membranes. Blots were probed with primary antibodies at 4 ºC overnight, and secondary antibody at room temperature for 1 h. Proteins of interest were visualized using Clarity Western ECL (Bio-Rad) and captured on ChemiDoc XRS$^+$ (Bio-Rad) using ImageLab software. Primary antibodies used were listed in Table S6.

**Real-time quantitative PCR**. Total RNA was extracted using Trizol reagent. cDNA was synthesized using PrimeScript RT reagent kit with gDNA eraser (Takara). qPCR was performed using TB Green Premix Ex Taq (Takara) in Quantstudio Flex 7 real-time PCR system (ThermoFisher). Primer sequences for qPCR were listed in Table S7.

**Chromatin immunoprecipitation (ChIP)-PCR**. Cells ($1 \times 10^7$) were crosslinked with 4% formaldehyde for 10 min, lysed in SDS lysis buffer (1% SDS, 10 mM EDTA, 50 mM Tris, pH 8.1) and sonicated to provide DNA fragments of ~200–400 bp. Pulldown was performed using anti-β-catenin (ab227499, Abcam, 2 µg/reaction), with rabbit IgG as the control. Immunoprecipitated chromatin was then purified using Protein G magnetic beads (Millipore), and DNA was reverse-crosslinked and purified. qPCR was performed using primers listed in Table S3.

**Cell viability assays**. Cell viability was determined by MTT assay. Colony formation was determined in 24-well plates seeded with 200–500 cells/well and incubated for 5–7 days. At the end point, the cells were stained with 0.2% crystal violet. Apoptosis was determined using Annexin-PE/7-aminoactinomycin D (7-AAD) staining kit (BD Biosciences, San Jose, CA) and flow cytometry. Cell cycle was analyzed by flow cytometry following ethanol fixation and PI staining. Compensation and algorithms for cell cycle analysis were generated using FlowJo software (v10) with default parameters. All flow cytometry was performed in a BD FACS Celesta machine. Gating strategies are shown in Fig. S18.

**Cell migration and invasion assays**. Confluent cultures in 12-well plates were scratched with sterile tips, washed with PBS and cultured in 1% FBS-supplemented MEM. Cells were photographed at 0, 24 and 48 h. Wound closure (%) was evaluated using the TScratch software. Cell invasion was determined using BioCoat Matrigel Invasion Chamber (BD Biosciences). Cells ($0.5-1.0 \times 10^5$) were seeded into the upper chamber in serum-free MEM. The lower chamber was filled with MEM with 10% FBS as a chemoattractant. After 72 h, cells invaded through the membrane were stained with crystal violet and counted.

**KRAS activity assay**. Active Ras Pull-Down and Detection Kit was used to determine KRAS status in cells. Active KRAS was detected using anti-KRAS antibody (ab180772) in the pulldown fractions.

**LDL-fluorescence assay**. For immunofluorescence assay, cells seeded on glass slides were incubated with BODIPY-labelled LDL (10 µg/mL, ThermoFisher) for

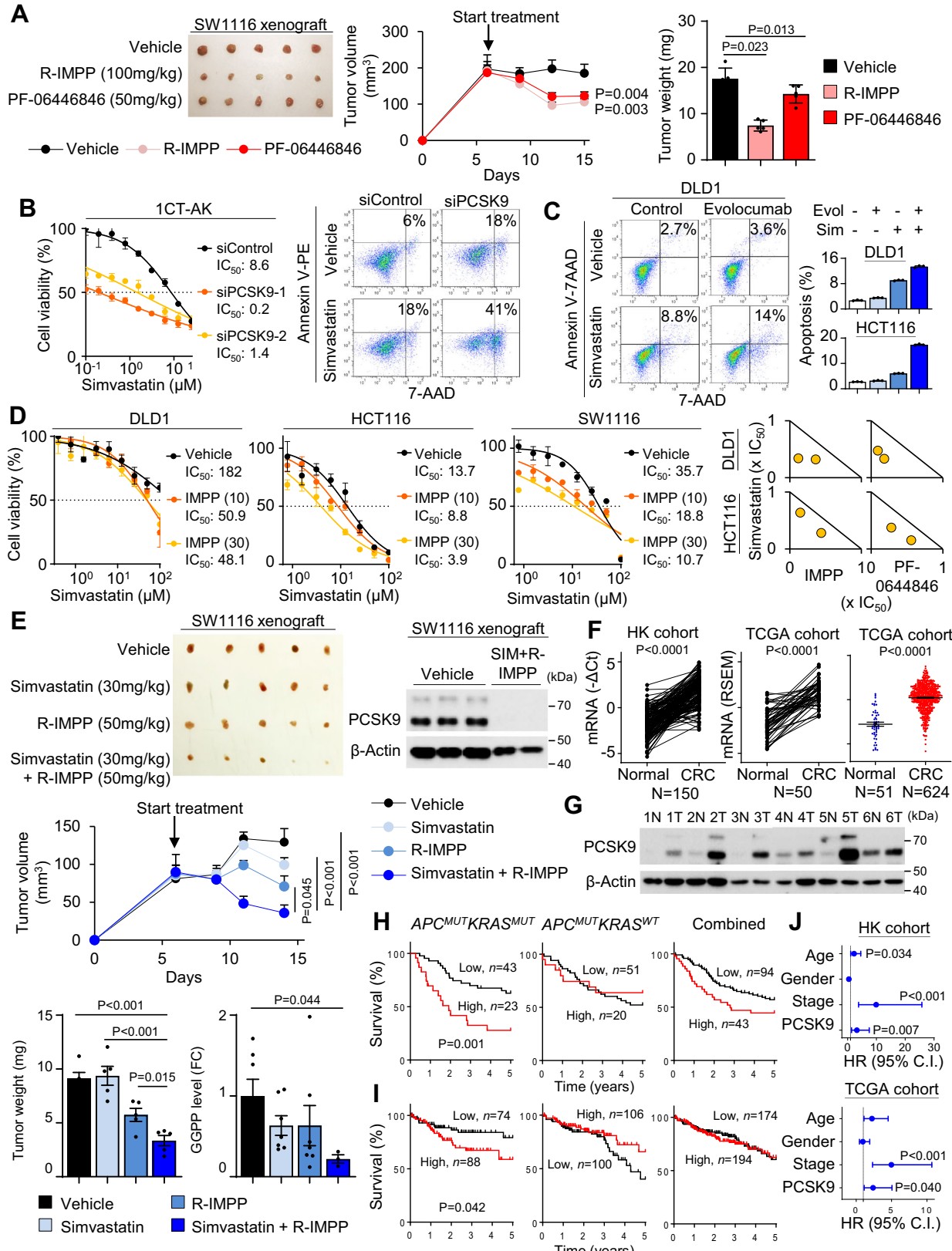

4 h at 37 ºC. Then, cells were fixed with 4% paraformaldehyde, loaded with DAPI-containing mounting medium and analyzed by confocal microscopy (Leica TCS SP8). For flow cytometry, cells also were incubated with BODIPY-labelled LDL for 4 h at 37 ºC, washed with PBS and harvested by trypsinization, and analyzed by flow cytometry. FITC channel was employed for the determination of BODIPY fluorescence.

**Statistics and reproducibility**. Data are presented as mean ± standard error (S.E.M.). In vitro studies were performed in at least triplicates in 2 independent experiments. All western blots were independently performed twice with consistent findings. Statistical analyses were performed with GraphPad Prism v6.0 or SPSS Statistics. Statistical comparisons between two groups were performed using two tailed, unpaired Student's t-test. Statistical comparisons for >2 groups were

**Fig. 6 PCSK9 and simvastatin synergistically inhibited *APC/KRAS*-mutant CRC growth and clinical significance of PCSK9 in CRC. A** Nude mice harboring SW1116 xenografts were treated with vehicle (PBS), R-IMPP (100 mg/kg), or PF-06446846 (50 mg/kg) ($n = 1$ experiment, $n = 5$ mice per group) by intraperitoneal injection every 2 days. Both R-IMPP and PF-06446846 inhibited tumor growth. **B** si*PCSK9* synergized with simvastatin to inhibit cell growth and induce apoptosis of 1CT-AK cells ($n = 3$, 72 h). **C** Evolocumab synergized with simvastatin to promote cell apoptosis in KRAS-mutant CRC cells (96 h). **D** The co-administration of R-IMPP or PF-0644846 reduced 48 h-$IC_{50}$ of simvastatin in KRAS-mutant CRC cells (*left*) ($n = 3$). Isobologram analyses indicated that combination of R-IMPP or PF-0644846 plus simvastatin were synergistic in suppressing cell viability (*right*). **E** Nude mice harboring SW1116 xenografts were treated with vehicle, R-IMPP, simvastatin, or their combination ($n = 1$ experiment, $n = 5$ mice per group). The combination drug treatment induced tumor regression and significantly suppressed tumor weight, PCSK9 protein expression and GGPP levels compared to vehicle or single treatment. **F** *PCSK9* mRNA is up-regulated in CRC tissues as compared with paired adjacent normal tissues in Hong Kong ($n = 150$ pairs), and TCGA cohorts ($n = 50$ pairs), and in unpaired samples from TCGA cohort ($n = 675$). **G** PCSK9 protein expression is increased in CRC compared to adjacent normal tissues assessed by western blot. **H** Tissue microarray cohort ($n = 137$) showed that PCSK9 protein expression predicts poor survival of $APC^{MUT}KRAS^{MUT}$ CRC ($n = 66$), but not in $APC^{MUT}KRAS^{WT}$ CRC ($n = 71$) or overall cohort. Log-rank test (two-tailed). **I** *PCSK9* mRNA predicts poor patient survival in $APC^{MUT}KRAS^{MUT}$ CRC ($n = 162$) in TCGA cohort, but not in $APC^{MUT}KRAS^{WT}$ CRC ($n = 206$) or overall cohort ($n = 368$). Log-rank test (two-tailed). **J** Multivariate COX proportional hazards regression analysis of the prognostic value of PCSK9 in $APC^{MUT}KRAS^{MUT}$ CRC patients in both Hong Kong and TCGA cohorts ($n = 162$, biological replicates). COX proportional hazards regression analysis. Data shown are means of biological replicates; ± SEM (**A–F**). Two-tailed Student's *t*-test (**A, E, F**). Two-tailed two-way ANOVA for growth curves (**A, E**). Source data are provided as a Source Data file.

performed using one-way analysis of variance (ANOVA) with Tukey post-hoc analysis. Statistical significance values were presented as exact values. Kaplan Meier curve was used to compare survival between patients with low or high PCSK9. Hazard ratios was performed with univariate and multivariate Cox regression models using SPSS software (IBM). A *p*-value < 0.05 is considered statistically significant.

**Reporting summary**. Further information on research design is available in the Nature Research Reporting Summary linked to this article.

## Data availability

Source data are provided with this paper. All TCGA dataset was accessed using UCSC Xena Browser (https://xenabrowser.net/). We have downloaded normalized RNA-seq and somatic mutation (SNP and INDEL) datasets directly from Xena Browser, from TCGA Colon and Rectal Cancer (COADREAD), GDC-TCGA Colon Cancer (COAD), GDC-TCGA Rectal Cancer (READ) cohort (Figs. 1 and 6). Correlation coefficients in Fig. S17 was derived from GEPIA (http://gepia.cancer-pku.cn/) based on TCGA COAD and TCGA READ cohorts. Source data are provided with this paper.

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

## Acknowledgements

This project was supported by funds from Research Grants Council-General Research Fund (24100520, C.C.W.; 14101917, C.C.W.; 14108718, C.C.W.), Research Grants Council-Collaborative Research Fund (C4039-19G, J.Y.); Heath and Medical Research Fund (06170686, C.C.W.; 08190706, C.C.W.); the Macau Science and Technology Development Fund (009/20176/A1, J.L.W.). Science and Technology Program Grant Shenzhen (JCYJ20170413161534162, J.Y.).

## Author contributions

CCW and JY designed and supervised the study. C.C.W., J.W., F.J., X.B., H.C., L.S.C., S.T.Y.L., S.T., J.X., Q.Z., D.L., H.S., H.G., and Z.C. generated the data for the study. W.K., A.H.K.C., and K.F.T. assisted with tissue microarray analysis. J.W.S. commented on the study.

## Competing interests

The authors declared no competing interests.
