## [Peer Review File · Nature Communications]

Title: The cholesterol uptake regulator PCSK9 promotes and is a therapeutic target in APC/KRAS-mutant colorectal cancerREVIEWER COMMENTS

Reviewer #1 (Remarks to the Author):

In this study, the authors explored the intracellular cholesterol metabolism in APC/KRAS mutant colorectal cancer and concluded that PCSK9 could promote KRAS activation through protein prenylation mediated by upregulated GGPP. First, the authors showed that LDL uptake was suppressed, and cholesterol biosynthesis was increased in APC/KRAS mutant cells. This metabolic alteration was accompanied by upregulated PCSK9 and the intermediate metabolite GGPP. Next, they show that PCSK9 induced cholesterol biosynthesis and GGPP accumulation, which promoted cell growth in APC/KRAS mutant CRC cells. Finally, they demonstrated that the genetic knockout or pharmacological inhibition of PCSK9 suppressed the growth of CRC organoids and xenografts. This work is generally well written and especially interesting in that cholesterol metabolism can be a potential therapeutic target in KRAS mutated CRC; however, a few issues need to be addressed.

1. The authors showed that the combination of statin and PCSK9 inhibition synergistically suppressed tumor growth in CRC xenografts. Did the authors observe any changes in GGPP amount among different treatments? This control is needed to support the synergistic effect of the combined treatment. PCSK9 levels in the xenografts undergoing the various treatments should also be informative.

2. Similarly, in the organoid experiments treated with different PKC inhibitors, transcriptional or protein data of PCSK9 and other molecules in the cholesterol biosynthesis cascade should be shown to demonstrate the effect of PCSK9 inhibition on cell viability and exclude the possibility of off-target effects.

3. Besides biosynthesis, intracellular cholesterol homeostasis is also maintained by uptake and efflux. In the intestinal epithelium, cholesterol uptake is mediated not only at the basal side by LDLR but also at the apical luminal side by NPC1L1. Likewise, cholesterol efflux is regulated by ABCA1 at the basal side and ABCG5 and ABCG8 at the apical side. Did the authors observe any alteration in these transporters in the APC and/or KRAS mutant organoids?

4. The authors claimed that the mechanism underlying PCSK9 upregulation in APC/KRAS mutant cells is caused by β -catenin activation. They showed that β -catenin knockdown decreased PCSK9 mRNA expression in 1CT-AK cells; however, the correlation between β -catenin and PCSK9 levels among 1CT, 1CT-K, 1CT-A, and 1CT-AK cells was not clear. To conclude that " β -catenin-induced PCSK9 represents a point of synergy between APC and KRAS," they need to show that the nuclear accumulation of β -catenin is more enriched in 1CT-AK than in 1CT-A cells in their setting.

5. In the same vein, the authors demonstrated that PCSK9 activated SREBP2; however, a previous report claimed that PCSK9 is a target of SREBP2 (DOI: 10.1074/jbc.M109.052407). Additionally, several molecules have been reported to regulate PCSK9, including E2F1 and FOXO3. Although a detailed analysis of each potential regulator might be out of the scope of this study, at least discussing the PCSK9

regulators and the difference between this study and other previous reports should benefit the readers to understand the complex regulatory mechanism better.

Minor points

1. In figure 1A, did the authors perform GSEA with gene sets of cholesterol homeostasis or biosynthesis? Using those gene sets would be a simple and convincing way to show that cholesterol biosynthesis was upregulated in APC/KRAS mutated CRC.
2. In the xenograft experiments, the authors should clearly indicate in the figure when the treatment was started (Figure 5A, C, Figure 6A, C).

Reviewer #2 (Remarks to the Author):

General comments:

Dr Yu and colleagues submit a comprehensive, interesting and timely manuscript discussing a metabolic vulnerability in Apc/Kras colorectal cancer. The authors should be congratulated on a clear, concise, well-constructed submission, I thoroughly enjoyed reading it. I have only a few minor suggestions to improve this manuscript, but in principle would certainly accept for publication.

Key results:

The manuscript demonstrates that, in comparison to KRAS WT cancers, APC/KRAS Mut/Mut cancers show distinct changes in lipid metabolism at a transcriptional level. They validate these changes in cell and animal models, and clearly show that this results in decreased lipid uptake, and enhanced de novo cholesterol synthesis. They elegantly demonstrate that this increase is to facilitate enhanced GGPP, promoting protein prenylation, which is facilitated through PCSK9. This axis contributes significantly to malignant phenotypes by enhancing MAPK signalling, and demonstrates an interesting target for pharmacological inhibition in APC/KRAS mut/mut cancers.

Validity:

The data is clearly presented, and very robust. The authors should be congratulated on such a thorough and comprehensive submission.

Significance:

The findings in this manuscript are significant. They show a tangible axis for targeting a hard-to-treat group of cancers. However, there are a number of papers in this sphere already, such as

<https://www.science.org/doi/abs/10.1126/scitranslmed.aaf8127>

<https://www.nature.com/articles/s41419-019-1544-y>

<https://cancerres.aacrjournals.org/content/canres/80/2/189.full.pdf>

It would be interesting to have the significance of the results further discussed in the context of what is already known/published, as it would strengthen the relevance of these unique findings.

Data and methodology:

The data presented are consistently of a high quality, and methods are appropriate to answer questions posed. A few more experimental details would be needed to ensure reproducibility and interpretation of results – such as timepoints at which samples are assessed (such as timepoints for siRNA, inhibitor treatment, assessment of death etc). This is particularly pertinent to ensure interpretation can be accurate – dead cells will not move, so if cells are dying or arrested when the motility assays have been run, this must be apparent so readers can account for this in their interpretation.

In addition, further details in methods on generation of cell lines (what lentiviral plasmid etc), data normalisation for metabolomics, formula for tumour volume calcs, vehicle details for treatment in vivo, flow analysis methods (compensation used, algorithms for cell cycle profiling) and number of technical replicates used are needed for clarity.

Analytical approach:

The analytical approaches taken within the manuscript are robust, appropriate and well considered. Further, the details supplied for these are sufficiently detailed.

Suggested improvements:

The manuscript is well constructed, data is comprehensive and sufficient to address the hypothesis posed. Furthermore, the conclusions are appropriate to what is presented.

There are only a few points that I think could improve the clarity of the manuscript, with little new data requested. Mostly it is changes in presentation order or wording to ensure conclusions are not over stretched.

1) The decrease in LDL could be accompanied with confirmed expression of LDLR early in the manuscript to consolidate this dataset.

2) The total cholesterol levels should be introduced at same time as the labelling of de novo cholesterol, so that this can be correctly interpreted.

3) On line 143, you discuss GGPP as “the most significantly induced metabolite in colon tumours compared to adjacent normal tissues” – firstly, here is this in relation only to these metabolites? The wording could be misleading, please clarify. Secondly, it has not been compared to Apc or Kras alone, and as such authors could comment on if these would also be different? Lastly, it is mentioned that this is seen alongside “moderate rise in GPP and FPP” (line 144), but the changes are noted as significant – these differences to the cell line data are not discussed.

4) Line 169 – PCSK9 is described as “the target of concurrent APC and KRAS mutations”, while it certainly is a target, it isn’t the only one, so this should be reworded

5) Line 246-247 reports that loss of PCSK9 decreases active KRAS, however it would be great to show the levels of KRAS activity across the isogenic panel of cell lines to show if it also increases as the levels of de novo cholesterol does? This is critical to support the proposed hypothesis, and would significantly strengthen data shown.

Minor issues: scale bar missing from Figure 1I; is should be are (line 76); Gene names should be italicised (eg. KRAS)

Clarity and context:

Data is presented in a clear and coherent way, with context outlined in introduction. A few points could be clarified as described in section above, to improve the context of conclusions drawn from data presented. Further consideration of published work could be added to the discussion, as already mentioned.

References:

References are relevant and appropriate.

Reviewer #3 (Remarks to the Author):

In this manuscript, the authors identified PCSK9 as a potential therapeutic target for APC/KRAS mutant CRC and explored the underlying mechanisms by which PCSK9 inhibitors could suppress colon tumorigenesis. PCSK9 expression was upregulated in APC/KRAS mutant CRC which increased de novo cholesterol biosynthesis and induces biosynthesis of a functional downstream metabolite GGPP, resulting in activation of the KRAS/MEK/ERK cascade in APC/KRAS mutant CRC to promote tumor cell growth. While the data are convincing, the following points need to be considered prior to publication.

1. Most of the mechanistic data are generated from genetically engineered APC/KRAS mutant CRC cell lines. The relevance to native primary colon tumors is lacking. This needs to be explored.
2. The mechanism by which PCSK9 regulates the de novo cholesterol pathway needs to be further validated. How does PCSK9 regulate GGPS1?
3. The authors consider GGPP as an oncometabolite. However, it has no pro-tumor effects in WT (sgControl) cancer cell lines (Figure 3H)?
4. Does genetic deficiency or pharmacological inhibition of GGPS1 show beneficial effects in vivo?
5. Does silencing or inhibition of the KRAS/MEK/ERK pathway abolish PCSK9-GGPP axis-induced colon cancer?
6. To demonstrate that PCSK9 inhibition has therapeutic effects on APC/KRAS mutant CRC in vivo, PCSK9 inhibitors should be administered to *ApcMin/+KrasG12D/+ Villin-Cre* mice to observe colon tumor growth differences between APC/KRAS-mutant and APC-mutant mice in which colon tumor development happens under a condition that more closely mimics human colon carcinogenesis than subcutaneous xenografts.
7. As a potential anti-tumor therapeutic strategy, PCSK9 inhibition should be able to kill tumor cells without affecting normal cells growth. How does PCSK9 inhibition affect normal cells growth and apoptosis? What are PCSK9 inhibitors' pharmacological effects on KRAS-WT human CRC organoids?
8. What are the plasma LDL and GGPP levels in mice implanted with sgPCSK9-transfected SW1116 and PCSK9 overexpressing DLD1 xenografts or treated with PCSK9 inhibitors?
9. In Figure 4D and 4E, the ratio of active and total KRAS expression need to be calculated to support the conclusion that PCSK9 overexpression/inhibition increased/decreased active KRAS expression since total KRAS expression was changed.
10. What is the effect of PCSK9 inhibition on colon tumors produced in the *Cdx2-Cre/Apc-floxed* mouse

model (PMID: 17942902)?

11. What percentage of human colon cancers are APC/KRAS double mutants?
12. In Figure 6, the authors analyzed the clinical samples to establish the correlation between PCSK9 and colon cancer. The GGPS1-GGPP pathway in human colon cancer samples need to measured?
13. The authors need to edit the manuscript to avoid mistakes such as no missing spaces between numbers and units, no dash between "24" and "well" for 24-well plates (Line 517) and such.
14. The authors need to proofread the manuscript. SREBP2 was written as SREBF2 in the legend to Figure 3.
15. Citations need to be inserted to Results section when a conclusion/finding was cited from other literatures such as Line 114-115, Line 185-186, Line 240-242.
16. Figure 1M: The tumor numbers should be noted in this experiment.
17. Figure 2E: Where is the cell viability time course for the LOVO cells?
18. Figure 2N: What is the expression of c-Myc in this experiment?
19. What is the % of APC/KRAS double mutants in humans?

Reviewer #1:

1. *The authors showed that the combination of statin and PCSK9 inhibition synergistically suppressed tumor growth in CRC xenografts. Did the authors observe any changes in GGPP amount among different treatments? This control is needed to support the synergistic effect of the combined treatment. PCSK9 levels in the xenografts undergoing the various treatments should also be informative.*

Response: We have now determined GGPP level in xenograft tumors treated with statin plus R-IMPP (**Figure 6E**), showing that the combination treatment significantly suppressed GGPP level, and is more effective in reducing GGPP compared to single drug treatment. PCSK9 expression in the combination treatment is also significantly reduced as compared to control (**Figure 6E**), which supports our hypothesis that GGPP as the target of combination statin and PCSK9 inhibitor treatment. This information has been added on **p.17, line 343-344**.

2. *Similarly, in the organoid experiments treated with different PKC inhibitors, transcriptional or protein data of PCSK9 and other molecules in the cholesterol biosynthesis cascade should be shown to demonstrate the effect of PCSK9 inhibition on cell viability and exclude the possibility of off-target effects.*

Response: We have now examined the effects of PCSK9 inhibitors on cholesterol pathway in the organoids model. As shown in **Figure 5J**, all three PCSK9 inhibitors suppressed mRNA levels of PCSK9, SREBF2, HMGCR and SQLE, the key enzymes involved in cholesterol biosynthesis pathway. Additionally, we have performed western blot to confirm the down-regulation of PCSK9 and HMGCR by PCSK9 inhibitors at the protein level (**Figure 5J**), further confirming effect of PCSK9 blockade on cholesterol biosynthesis pathway. This information has been added on **p.15, line 311-312**.

3. *Besides biosynthesis, intracellular cholesterol homeostasis is also maintained by uptake and efflux. In the intestinal epithelium, cholesterol uptake is mediated not only at the basal side by LDLR but also at the apical luminal side by NPCL11. Likewise, cholesterol efflux is regulated by ABCA1 at the basal side and ABCG5 and ABCG8 at the apical side. Did the authors observe any alteration in these transporters in the APC and/or KRAS mutant organoids?*

Response: Thank you for your suggestion. We have now performed qPCR of ABCA1, ABCG5, ABCG8 and NPCL11 in 1CT, 1CT-K, 1CT-A and 1CT-AK cells. As shown in **Figure S2**, ABCA1 is up-regulated in 1CT-K and 1CT-AK cells; whereas NPCL11 is down-regulated in 1CT-A and 1CT-AK cells. ABCG5 and ABCG8 mRNA is undetectable. Hence, APC loss plus mutant KRAS also increased cholesterol efflux (ABCA1) and decreased cholesterol uptake (NPCL11). This may also contribute to increased cholesterol uptake. PCSK9 overexpression or knockout has no consistent effect on the expression of ABCA1, ABCG5, ABCG8 and NPCL11,

suggesting that the oncogenic effect of PCSK9 is independent of these genes (**Figure S2**). In organoids, PCSK9 inhibitors did not show consistent effect with respect to ABCA1 expression (**Figure S2**), while rest of the genes are not detectable. This information has been added to **p.7, line 124-127, p.11, line 222-224, and p.15, line 312-313**.

4. The authors claimed that the mechanism underlying PCSK9 upregulation in APC/KRAS mutant cells is caused by b-catenin activation. They showed that β -catenin knockdown decreased PCSK9 mRNA expression in ICT-AK cells; however, the correlation between β -catenin and PCSK9 levels among ICT, ICT-K, ICT-A, and ICT-AK cells was not clear. To conclude that “ β -catenin-induced PCSK9 represents a point of synergy between APC and KRAS,” they need to show that the nuclear accumulation of b-catenin is more enriched in ICT-AK than in ICT-A cells in their setting.

Response: We have now performed western blot analysis of active β -catenin in ICT isogenic cell lines. ICT-AK cells showed increased nuclear translocation of active β -catenin, increased TOPflash activity and higher expression of multiple WNT downstream targets (Cyclin D1/D2, CD44, EPHB2 and c-MYC) (**Figure S5**). These data collectively imply that β -catenin activity is induced in ICT-AK cells compared to ICT-A cells. This information has been added to **p.10, line 188-190**.

5. In the same vein, the authors demonstrated that PCSK9 activated SREBP2; however, a previous report claimed that PCSK9 is a target of SREBP2 (DOI: 10.1074/jbc.M109.052407). Additionally, several molecules have been reported to regulated PCSK9, including E2F1 and FOXO3. Although a detailed analysis of each potential regulator might be out of the scope of this study, at least discussing the PCSK9 regulators and the difference between this study and other previous reports should benefit the readers to understand the complex regulatory mechanism better.

Response: We have now provided a discussion of PCSK9 regulators and its role in CRC, as follow: “PCSK9 expression is also positively regulated by SREBP2 and E2F1,^{23,24} while being repressed by FOXO3.²⁵ Since PCSK9 up-regulated SREBP2 in CRC cells, β -catenin-driven PCSK9 might drive a positive feedback loop involving PCSK9 and SREBP2.” This has been added to **p.20, line 419-422**.

Minor points

1. In figure 1A, did the authors perform GSEA with gene sets of cholesterol homeostasis or biosynthesis? Using those gene sets would be a simple and convincing way to show that cholesterol biosynthesis was upregulated in APC/KRAS mutated CRC.

Response: We have performed GSEA analysis using KEGG database. In this database, the gene set for cholesterol metabolism (K04979) mainly consists of genes involved in lipoprotein

transport, and is not enriched by our analysis. De novo biosynthesis pathway is incorporated into 2 gene sets: terpenoid biosynthesis (mevalonate pathway) and steroid biosynthesis (distal cholesterol synthesis) pathways, that are both enriched in our analysis.

2. *In the xenograft experiments, the authors should clearly indicate in the figure when the treatment was started (Figure 5A, C, Figure 6A, C).*

Response: We have now indicated the treatment time in **Figure 6A** and **6C**. **Figure 5A** and **5C** are based on PCSK9 overexpression or knockout in CRC cells and no treatment is given.

Reviewer #2:

Significance:

The findings in this manuscript are significant. They show a tangible axis for targeting a hard-to-treat group of cancers. However, there are a number of papers in this sphere already, such as <https://www.science.org/doi/abs/10.1126/scitranslmed.aaf8127>

<https://www.nature.com/articles/s41419-019-1544-y>

<https://cancerres.aacrjournals.org/content/canres/80/2/189.full.pdf>

It would be interesting to have the significance of the results further discussed in the context of what is already known/published, as it would strengthen the relevance of these unique findings.

Response: We have now discussed the articles in context with our data as follows: “Mevalonate pathway blockers such as statins and Tasin-1 have been shown to suppress glycolysis³⁰, pyrimidine bioynthesis³¹, and induce cell death in cancer cells. By combining statin and PCSK9 inhibitors, a more effective suppression of mevalonate pathway could be achieved, thereby further improving therapeutic efficacy”. This has been added to **p.21, line 455-458**.

Data and methodology:

The data presented are consistently of a high quality, and methods are appropriate to answer questions posed. A few more experimental details would be needed to ensure reproducibility and interpretation of results – such as timepoints at which samples are assessed (such as timepoints for siRNA, inhibitor treatment, assessment of death etc). This is particularly pertinent to ensure interpretation can be accurate – dead cells will not move, so if cells are dying or arrested when the motility assays have been run, this must be apparent so readers can account for this in their interpretation. In addition, further details in methods on generation of cell lines (what lentiviral plasmid etc), data normalisation for metabolomics, formula for tumour volume calcs, vehicle details for treatment in vivo, flow analysis methods (compensation used, algorithms for cell cycle profiling) and number of technical replicates used are needed for clarity.

Response: We have added additional information in the methods section. Gating strategies for flow cytometry are now added on **Figure S18**. We have also provided all source data for the

paper and added the number of technical replicates corresponding figure legends.

There are only a few points that I think could improve the clarity of the manuscript, with little new data requested. Mostly it is changes in presentation order or wording to ensure conclusions are not over stretched.

1) The decrease in LDL could be accompanied with confirmed expression of LDLR early in the manuscript to consolidate this dataset.

Response: We have moved **Figure 1K** to **Figure 1E** accordingly.

2) The total cholesterol levels should be introduced at same time as the labelling of de novo cholesterol, so that this can be correctly interpreted.

Response: We have moved the total cholesterol levels to **Figure 1F**.

3) On line 143, you discuss GGPP as “the most significantly induced metabolite in colon tumours compared to adjacent normal tissues” – firstly, here is this in relation only to these metabolites? The wording could be misleading, please clarify. Secondly, it has not been compared to Apc or Kras alone, and as such authors could comment on if these would also be different? Lastly, it is mentioned that this is seen alongside “moderate rise in GPP and FPP” (line 144), but the changes are noted as significant – these differences to the cell line data are not discussed.

Response: We have reworded this sentence, and commented on the potential effects of single Apc or Kras mutation on GGPP, as follows: “As shown in **Figure 1I**, GGPP ($P < 0.01$), geranyl pyrophosphate (GPP) ($P < 0.05$), and farnesyl pyrophosphate (FPP) ($P < 0.05$) are up-regulated in colon tumors compared to adjacent normal tissues (**Figure 1I**), implying higher isoprenoid biosynthesis, especially GGPP. We speculated that GGPP might also be up-regulated *Apc^{Min/+}* given its increased level in 1CT-A cells. Concomitant up-regulation of GPP, FPP and GGPP may imply more robust up-regulation of mevalonate pathway in mice as compared to cell line models.” This has been added to **p.8, line 147-153**.

4) Line 169 – PCSK9 is described as “the target of concurrent APC and KRAS mutations”, while it certainly is a target, it isn’t the only one, so this should be reworded

Response: We have now reworded this sentence to “These data suggest that PCSK9 expression is driven by concurrent *APC* and *KRAS* mutations in CRC.”. This has been added to **p.9, line 176-177**.

5) Line 246-247 reports that loss of PCSK9 decreases active KRAS, however it would be great to show the levels of KRAS activity across the isogenic panel of cell lines to show if it also increases as the levels of de novo cholesterol does? This is critical to support the proposed

hypothesis, and would significantly strengthen data shown.

Response: We have now compared the active KRAS in 1CT-A and 1CT-AK isogenic cell, with or without PCSK9 knockdown. As shown in **Figure S9**, 1CT-AK control cells (with induced cholesterol pathway) exhibit higher levels of active KRAS compared to 1CT-A control cells. Interestingly, knockdown of PCSK9 decreases the active KRAS in 1CT-AK cells, but has no corresponding effect on 1CT-A cells. These data support our hypothesis that activated de novo cholesterol synthesis is associated with increased active KRAS activity. This has been added to **p.13, line 261-264**.

Minor issues: scale bar missing from Figure 11; is should be are (line 76); Gene names should be italicised (eg. KRAS)

Response: We have addressed these points accordingly.

Clarity and context:

Data is presented in a clear and coherent way, with context outlined in introduction. A few points could be clarified as described in section above, to improve the context of conclusions drawn from data presented. Further consideration of published work could be added to the discussion, as already mentioned.

Response: We have provided additional discussion on **p.21, line 455-458**.

Reviewer #3:

1. Most of the mechanistic data are generated from genetically engineered APC/KRAS mutant CRC cell lines. The relevance to native primary colon tumors is lacking. This needs to be explored.

Response: To establish the correlation of PCSK9 to cholesterol biosynthesis and MEK-ERK signalling in CRC patients, we have examined the correlations between PCSK9 and these genes in CRC tumor tissues from TCGA CRC cohort. As shown in **Figure S17**, PCSK9 expression positively correlated with *SREBP2*, *HMGCR*, *FDFT1*, *SQLE* (cholesterol biosynthesis) and *CCND1*, *CCNE1* (MEK-ERK). The results in CRC patients are largely in accordance with our mechanistic findings. This has been added **p.18, line 373-379**.

2. The mechanism by which PCSK9 regulates the de novo cholesterol pathway needs to be further validated. How does PCSK9 regulate GGPS1?

Response: We showed that PCSK9 positively regulated SREBP2, an important transcription factor for *de novo* cholesterol pathway. We therefore performed siRNA-mediated knockdown of SREBP2. Indeed, knockdown of SREBP2 abolished effect of PCSK9 overexpression on the induction of cholesterol biosynthesis genes (**Figure S7**), suggesting that PCSK9 regulates *de novo* cholesterol biosynthesis via SREBP2. Interestingly, PCSK9 also positively regulated the

expression of GGPS1 (**Figure S7**), and such an effect is dependent on SREBP2 (**Figure S7**). This information has been added to **p.12, line 227-229** and **line 238**.

3. *The authors consider GGPP as an oncometabolite. However, it has no pro-tumor effects in WT (sgControl) cancer cell lines (Figure 3H)?*

Response: While GGPP is a potential oncometabolite, we observed that in SW1116 and LOVO control cells, GGPP only increased cell growth in the context of PCSK9 knockout. It is probable the endogenous GGPP production in these control cells is already sufficient for mutant KRAS activation, and the addition of exogenous GGPP thus had no further effect.

4. *Does genetic deficiency or pharmacological inhibition of GGPS1 show beneficial effects in vivo?*

Response: Thank you for your suggestion. We have performed GGPS1 knockdown in SW1116 and LOVO cells with high PCSK9 expression. GGPS1 knockdown inhibits cell proliferation *in vitro* and the growth of SW1116 xenografts *in vivo* (**Figure S8**). This information has been added to **p.12, line 246-248**.

5. *Does silencing or inhibition of the KRAS/MEK/ERK pathway abolish PCSK9-GGPP axis-induced colon cancer?*

Response: We have performed additional experiments with DLD1-vector and DLD1 PCSK9-overexpressing cells with MEK1/2 inhibitor AZD6244. As shown in **Figure S11**, treatment with AZD6244 inhibitor abrogated PCSK9 overexpression-induced cell viability in DLD1 cells after 48h. IC₅₀ values (72h) showed that DLD1 cells with PCSK9 overexpression is more sensitive (IC₅₀: 10.8µM) to AZD6244 compared to control cells (IC₅₀: 40.5µM), consistent with the role of KRAS/MEK/ERK signalling downstream of PCSK9-GGPP axis. This information has been added to **p.14, line 278-280**.

6. *To demonstrate that PCSK9 inhibition has therapeutic effects on APC/KRAS mutant CRC in vivo, PCSK9 inhibitors should be administered to Apc^{Min/+}Kras^{G12D/+}Villin-Cre mice to observe colon tumor growth differences between APC/KRAS-mutant and APC-mutant mice in which colon tumor development happens under a condition that more closely mimics human colon carcinogenesis than subcutaneous xenografts.*

Response: To investigate the role of PCSK9 inhibition on spontaneous CRC, we have treated *Apc^{Min/+}Kras^{G12D/+}Villin-Cre* mice with R-IMPP (50mg/kg, i.p., 3 times/week). As shown in **Figure S16**, R-IMPP significantly suppressed tumor number ($P<0.01$) and tumor load ($P<0.01$) compared to control, validating the efficacy of PCSK9 blockade in APC/KRAS-mutant CRC. This information has been added to **p.16, line 320-323**.

7. As a potential anti-tumor therapeutic strategy, PCSK9 inhibition should be able to kill tumor cells without affecting normal cells growth. How does PCSK9 inhibition affect normal cells growth and apoptosis? What are PCSK9 inhibitors' pharmacological effects on KRAS-WT human CRC organoids?

Response: We have performed cell viability (MTT) assay to assess effect of PCSK9 inhibition on normal colon epithelial cells, NCM460. 72h-IC₅₀ values for R-IMPP and PF-0644846 were 57.5 and >100μM, respectively, which are >2 and >10-fold higher compared to KRAS-mutant CRC cells (**Figure 5H**). Both drugs do not induce apoptosis in NCM460 cells (**Figure S14**). We determine the effect of Evolocumab, R-IMPP and PF-0644846 in KRAS-wildtype CRC organoid (POP818, University of Toronto), showing that PCSK9 inhibition has no effect on KRAS-wildtype CRC organoids (**Figure S15**). This information has been added to **p.15, line 308-309** and **line 313-314**.

8. What are the plasma LDL and GGPP levels in mice implanted with sgPCSK9-transfected SW1116 and PCSK9 overexpressing DLD1 xenografts or treated with PCSK9 inhibitors?

Response: We have now evaluated GGPP levels DLD1 xenografts with PCSK9 overexpression and SW1116 xenografts undergoing drug treatment. Consistent with our hypothesis, PCSK9-overexpressing tumors have elevated GGPP ($P<0.01$); whereas PCSK9 knockout suppresses GGPP levels in SW1116 xenografts *in vivo* ($P<0.05$) (**Figure 5F**). We have also measured the levels of LDL in mice bearing SW1116 control and sgPCSK9 xenografts, but observed no difference between the two groups. This information has been added to **p.14, line 293-296**.

9. In Figure 4D and 4E, the ratio of active and total KRAS expression need to be calculated to support the conclusion that PCSK9 overexpression/inhibition increased/decreased active KRAS expression since total KRAS expression was changed.

Response: We have now analyzed the ration of active to total KRAS by densitometry analysis (**Figure 4D and 4E**), confirming that active KRAS expression is increased or decreased by PCSK9 overexpression and inhibition, respectively.

10. What is the effect of PCSK9 inhibition on colon tumors produced in the Cdx2-Cre/Apc-floxed mouse model (PMID: 17942902)?

Response: We have performed R-IMPP treatment in Apc^{Min/+}Kras^{G12D/+}Villin-Cre mice (Q6) as a model of spontaneous CRC. However, we are unable to perform the same treatment on this particular model as it is not available at our hands.

11. What percentage of human colon cancers are APC/KRAS double mutants?

Response: In TCGA CRC cohort, APC/KRAS double mutants account for 34.1% (182/534) of CRC patients. In our TMA cohort, APC/KRAS double mutants are present in 36.7% (66/180)

of CRC patients. This information has been added to discussion section on **p.22, line 460-461**.

12. In Figure 6, the authors analyzed the clinical samples to establish the correlation between PCSK9 and colon cancer. The GGPS1-GGPP pathway in human colon cancer samples need to measured?

Response: We have now performed additional work to investigate the clinical significance of GGPS1 in CRCs. In TCGA cohort, GGPS1 is up-regulated in CRC tumor tissues compared to adjacent normal tissues at mRNA level (**Figure S17**). GGPS1 expression is positively correlated to PCSK9 in TCGA and HK cohorts (**Figure S17**). Moreover, GGPS1 protein predicts poor survival of KRAS-mutant CRC ($P=0.05$), but not KRAS-wildtype CRC in our HK cohort (**Figure S17**); a similar trend is observed in TCGA cohort (**Figure S17**). Our data collectively imply that GGPS1 is up-regulated in CRC, correlates with PCSK9 expression and predicts poor survival in KRAS-mutant CRC patients. This information has been added to **p.18, line 373-379**.

13. The authors need to edit the manuscript to avoid mistakes such as no missing spaces between numbers and units, no dash between “24” and “well” for 24-well plates (Line 517) and such.

Response: Revised as suggested.

14. The authors need to proofread the manuscript. SREBP2 was written as SREBF2 in the legend to Figure 3.

Response: Revised as suggested.

15. Citations need to be inserted to Results section when a conclusion/finding was cited from other literatures such as Line 114-115, Line 185-186, Line 240-242.

Response: Citations have been added.

16. Figure 1M: The tumor numbers should be noted in this experiment.

Response: This analysis consists of 5 adjacent normal and 6 CRC samples. We have now added this information to the figure legends.

17. Figure 2E: Where is the cell viability time course for the LOVO cells?

Response: We have now added this data to **Figure 2E**.

18. Figure 2N: What is the expression of c-Myc in this experiment?

Response: We have performed western blot, and confirmed up-regulation of c-Myc.

19. What is the % of APC/KRAS double mutants in humans?

Response: In TCGA Pan-Cancer cohort, APC/KRAS double mutants only account for 2.2% of (230/10429) all cancers. Hence, the majority of APC/KRAS double mutants (79.1%, 182/230) are CRC.

REVIEWER COMMENTS

Reviewer #1 (Remarks to the Author):

The authors have fully addressed my concerns.

Reviewer #3 (Remarks to the Author):

We have no additional concerns

Reviewer #4 (Remarks to the Author):

Cholesterol is a key component in the regulation of various cellular processes under physiological and pathological conditions. In particular it has been shown that cholesterol biosynthesis is required for cell viability. This is mediated by the induction of several intracellular signaling cascades, although the mechanism involved in the cholesterol induced carcinogenesis of APC/KRAS-mutant colorectal cancer is not fully understood. Here the authors studied the cholesterol homeostasis in this colorectal cancer, and found that de-novo cholesterol biosynthesis is indeed induced in APC/KRAS-mutant colorectal cancer.

This is accompanied by an increased geranylgeranyl diphosphate (GGPP) level, and induction of the cholesterol-related PCSK9. Mechanistically, PCSK9 reduced cholesterol uptake, which in turn, promotes de novo cholesterol and GGPP biosynthesis and induced the ERK cascade that may be related to survival. Interestingly, PCSK9 inhibitors suppressed the growth of APC/KRAS mutant colorectal cancer and PCSK9 overexpression predicted poor survival in this type of cancer. Therefore, PCSK9 can serve as a therapeutic target and prognostic factor for APC/KRAS mutant colorectal cancer.

Overall, this is a very comprehensive interesting paper that sheds a new light on the mechanism by which cholesterol biosynthesis is involved in the viability of APC/KRAS mutant colorectal cancer. The amount of work invested in this paper is really impressive. The experiments are usually well performed, with mostly significant results, and the authors adequately addressed most concerns raised by reviewer 2. Unfortunately however, some of the blot results are still not convincing. These, as well as a few other minor points listed below should be addressed as follows:

1. A main concern I have is the blot results that are not always convincing. I strongly recommend correcting them, and particularly include quantification and statistical analysis of most of the blots.
2. In particular, the results obtained with the anti pMEK, MEK, pERK and ERK antibodies are somewhat strange. First, the results obtained with the pMEK and pERK are not always similar to each other as expected. Second, the appearance of the ERKs and MEKs is different between the different blots (one isoform in some but two isoforms in others, see also Fig. S10) and the gaps between the two bands appear in some of the ERK blots is not consistent (particularly in 4F in SW1116 cells and in 4G). One reason for these changes might be the use of "basal state" cell, in which the phosphorylation is very low. Is it possible to show the results of stimulated cells? Changes in pAKT can validate the MEK/ERK results

as well. Finally, the quantification and statistical analysis of the data (as suggested above) may address this point as well.

3) In Fig. 2N, the results are not convincing.

4) In Fig. 4C, it is not clear to me why there is no KRas is not found in the cytoplasm under any condition.

5) In the title in line 159 is somewhat confusing, and “induces” should be changed to induce.

6) Line 204, change to “invasion and migration” rather than “migration and invasion” to fit the figure numbers.

Response to Reviewer #4:

1. A main concern I have is the blot results that are not always convincing. I strongly recommend correcting them, and particularly include quantification and statistical analysis of most of the blots.

Response: We have now provided quantification for all the blots (**all main Figures**), as well as statistical analysis where appropriate (**Figure S19**). We have also revised **Fig. 2N, 4F, 4G, 4H, and S10** according to your suggestion.

2. In particular, the results obtained with the anti pMEK, MEK, pERK and ERK antibodies are somewhat strange. First, the results obtained with the pMEK and pERK are not always similar to each other as expected. Second, the appearance of the ERKs and MEKs is different between the different blots (one isoform in some but two isoforms in others, see also Fig. S10) and the gaps between the two bands appear in some of the ERK blots is not consistent (particularly in 4F in SW1116 cells and in 4G). One reason for these changes might be the use of “basal state” cell, in which the phosphorylation is very low. Is it possible to show the results of stimulated cells? Changes in pAKT can validate the MEK/ERK results as well. Finally, the quantification and statistical analysis of the data (as suggested above) may address this point as well.

Response: Thanks for your insightful comments. From our experience, ERK/p-ERK consists of 2 bands (p42/p44) and MEK/p-MEK consists of 1 band, but generally the band for p-ERK (p44) was weak and was not always exposed. We have now replaced the blots for p-ERK/ERK for SW1116 cells in **Fig. 4F** and DLD1 cells in **Fig. 4G** with consistent gaps. p-ERK blots in **Fig. S10** and **Fig. 4H** have been replaced with those with longer exposure to reveal p44 ERK; p-MEK for SW1116 have been repeated for **Fig. S10**. We performed additional western blot of p-AKT in **Fig. 4F** and **4G**, showing that p-AKT was down-regulated by PCSK9 knockout in SW1116 cells; whilst PCSK9 overexpression in DLD1 induced p-AKT. Finally, we have now performed stimulation in SW1116-sgControl and SW1116-sgPCSK9 cells using EGF (10ng/mL, 30min). As shown in **Figure S9**, both p-MEK and p-ERK (p42/p44) was detected after the addition of EGF, with reduced expression in PCSK9 knockout cells. This information has been added to **p.13, line 271-272**.

3) In Fig. 2N, the results are not convincing.

Response: We have now repeated the western blot with improved data for **Fig. 2N**.

4) In Fig. 4C, it is not clear to me why there is no KRas is not found in the cytoplasm under any condition.

Response: We have performed this western blot for multiple times, and we did observe that Kras is predominantly expressed in the membrane fraction. We can detect Kras in cytosolic fraction over an extend exposure period, indicating that cytosolic Kras is very lowly expressed. This might be attributed to the fact that we use mutant Kras cell lines, and thus have more robust Kras activation and membrane localization.

5) In the title in line 159 is somewhat confusing, and “induces” should be changed to induce.

Response: Revised as suggested.

6) Line 204, change to “invasion and migration” rather than “migration and invasion” to fit the figure numbers.

Response: Revised as suggested.

REVIEWERS' COMMENTS

Reviewer #4 (Remarks to the Author):

The authors successfully addressed all my points. I have no further comments

Response to reviewer's comments:

Reviewer #4 (Remarks to the Author):

The authors successfully addressed all my points. I have no further comments

Response: We thank you for the positive comments.